# Additive effects on the energy barrier for synaptic vesicle fusion cause supralinear effects on the vesicle fusion rate

Sebastiaan Schotten[1†], Marieke Meijer[1†], Alexander Matthias Walter[1,2,3†], Vincent Huson[1], Lauren Mamer[4], Lawrence Kalogreades[1], Mirelle ter Veer[1], Marvin Ruiter[1], Nils Brose[5], Christian Rosenmund[4], Jakob Balslev Sørensen[2‡], Matthijs Verhage[1‡], Lennart Niels Cornelisse[1‡*]

[1]Department of Functional Genomics, Center for Neurogenomics and Cognitive Research, VU University Medical Center, Amsterdam, Netherlands; [2]Department of Neuroscience and Pharmacology, Faculty of Health Sciences, University of Copenhagen, Copenhagen, Denmark; [3]Molecular Physiology and Cell Biology, Leibniz Institute for Molecular Pharmacology, Berlin, Germany; [4]NeuroCure Cluster of Excellence, Neuroscience Research Center, Charité - Universitätsmedizin Berlin, Berlin, Germany; [5]Department of Molecular Neurobiology, Max Planck Institute for Experimental Medicine, Göttingen, Germany

**\*For correspondence:**
l.n.cornelisse@vu.nl

[†]These authors contributed equally to this work

[‡]These authors also contributed equally to this work

**Competing interests:**
See page 21

**Abstract** The energy required to fuse synaptic vesicles with the plasma membrane ('activation energy') is considered a major determinant in synaptic efficacy. From reaction rate theory, we predict that a class of modulations exists, which utilize linear modulation of the energy barrier for fusion to achieve supralinear effects on the fusion rate. To test this prediction experimentally, we developed a method to assess the number of releasable vesicles, rate constants for vesicle priming, unpriming, and fusion, and the activation energy for fusion by fitting a vesicle state model to synaptic responses induced by hypertonic solutions. We show that complexinI/II deficiency or phorbol ester stimulation indeed affects responses to hypertonic solution in a supralinear manner. An additive vs multiplicative relationship between activation energy and fusion rate provides a novel explanation for previously observed non-linear effects of genetic/pharmacological perturbations on synaptic transmission and a novel interpretation of the cooperative nature of $Ca^{2+}$-dependent release.

## Introduction

Regulation of synaptic efficacy is an essential aspect of information processing in neuronal networks. The energy barrier for vesicle fusion is considered to be a main contributing factor. To release neurotransmitters, synaptic vesicles (SVs) need to fuse with the neuronal plasma membrane, which requires substantial energy. Local membrane deformation, dehydration of lipid head groups, neutralization of opposite membrane charges, lipid splaying, and the creation of a lipid stalk all contribute to the energy barrier that needs to be overcome before neurotransmitters are released (*Kuzmin et al., 2001*; *Kozlovsky and Kozlov, 2002*; *Markin and Albanesi, 2002*; *Sorensen, 2009*). Reaction rate theory suggests that specifically modulation of the fusion energy barrier is a powerful way to regulate synaptic efficacy. According to the Arrhenius equation, reaction rates change exponentially with changes in the activation energy, which is the minimum energy required for a reaction (e.g., vesicle fusion) (*Arrhenius, 1889a, 1889b*). Thus, we predict that a set of modulations of the release rate may exist, which act by lowering the activation energy for fusion. If this is the case, they will have a supra-linear effect on the

**eLife digest** Information is carried around our nervous system by cells called neurons, which are connected to each other by junctions known as synapses. Within the neurons, there are many small compartments known as synaptic vesicles that are essential to the transfer of information from one neuron to the next. When one neuron is activated, the synaptic vesicles fuse with the membrane surrounding the cell to release molecules called neurotransmitters, which cross the synapse and activate the next neuron. Vesicle fusion is carefully regulated to control the speed and amount of neurotransmitter release, which determines the level of activation of the next neuron.

Vesicle fusion requires energy, much of which is provided by a set of proteins found in the synapse. The minimum amount of energy required—called the activation energy—is influenced by many factors, including the shape of the cell's membrane at the synapse. It is thought that altering the activation energy required for fusion may control the activity of synapses, but it is not possible to directly measure this in living cells.

To bypass this problem, Schotten, Meijer, Walter et al. established a new method to study vesicle fusion. This method combines a mathematical model with experimental data of the activity of synapses. First, the neurons were placed in a solution containing the sugar sucrose, which triggered vesicle fusion by lowering the activation energy. The increase in vesicle fusion was smaller in neurons that lacked two proteins called complexin I and complexin II—which control vesicle fusion—than in the normal neurons.

A molecule called phorbol ester is also able to activate the release of neurotransmitters. When cells were treated with both sucrose and phorbol ester, the speed of vesicle fusion was greater. The experiments show that the effects of sucrose, phorbol ester, and the complexins multiply together to dramatically alter vesicle fusion.

Schotten, Meijer, Walter et al. suggest a new model for how the activation energy of vesicle fusion controls the transfer of information across synapses. This might shed new light on how the efficiency of vesicle fusion is altered when neurons are highly active, which is thought to have strong implications for how information is processed in the brain.

fusion rates, and converting rates to energies (by inverting the Arrhenius equation) should reveal additive effects on the fusion barrier. This is highly relevant since many presynaptic factors may act on the activation energy for fusion simultaneously and potentially independently during synaptic stimulation.

Much of the energy required for SV fusion is likely provided by the SNARE proteins, synaptobrevin/ VAMP, syntaxin, and SNAP25, whose assembly into a trimeric SNARE-complex drives the fusion reaction (*Sorensen, 2009*; *Jahn and Fasshauer, 2012*). However, several other proteins likely contribute to the efficient and fast reduction of the activation energy for SV fusion that is required for fast synaptic transmission. During action potential (AP) stimulation, for example, SV fusion rates increase by several orders of magnitude within a few milliseconds due to the rapid activation of $Ca^{2+}$ sensors of the synaptotagmin-family, which control SNARE-mediated fusion (*Rhee et al., 2005*; *Xu et al., 2007*; *Walter et al., 2010*; *Weber et al., 2010*; *Kochubey and Schneggenburger, 2011*; *Arancillo et al., 2013*; *Sudhof, 2013*). Other proteins, such as Munc18 and Munc13, might also support synaptic transmission by reducing the activation energy for SV fusion, either through their established roles in SNARE-complex assembly (*Basu et al., 2007*; *Wierda et al., 2007*; *Weber et al., 2010*; *Xue et al., 2010*; *Arancillo et al., 2013*) or through independent actions.

Direct measurements of the exact contributions of different molecular events inside living nerve terminals to the activation energy for SV fusion are not possible. However, the predicted supra-linear modulation of release rates can be measured experimentally. This can be interpreted as changes in the activation energy under certain assumptions (e.g., a constant empirical prefactor A, see below). SV release kinetics has been intensively studied using flash photolysis of caged $Ca^{2+}$ (*Schneggenburger and Neher, 2000*; *Lou et al., 2005*; *Sakaba et al., 2005*; *Korogod et al., 2007*; *Sun et al., 2007*; *Wolfel et al., 2007*; *Kochubey and Schneggenburger, 2011*; *Burgalossi et al., 2012*). However, synaptic responses to $Ca^{2+}$ elevation (either triggered by natural stimulations by APs or by $Ca^{2+}$ uncaging) are caused by a rapid synaptotagmin/$Ca^{2+}$-induced lowering of the energy barrier for vesicle fusion. This mechanism might be modified by several factors that interact with synaptotagmin.

Therefore, to assess changes in the energy barrier per se, caused by other factors, we must use a different, $Ca^{2+}$-independent method to assess changes in release kinetics. In this regard, hypertonic solutions have been used widely as they cause SV release from the same readily releasable SV pool (RRP) as APs do, but by a $Ca^{2+}$-independent stimulus (*Fatt and Katz, 1952*; *Stevens and Tsujimoto, 1995*; *Rosenmund and Stevens, 1996*). Correspondingly, changes in the kinetics of synaptic responses to hypertonicity-induced SV fusion have been interpreted as changes in the intrinsic 'release willingness' or 'fusogenicity' of SVs, which may represent an inverse measure for the activation energy for SV fusion (*Basu et al., 2007*; *Wierda et al., 2007*).

Here, we introduce a method to quantify vesicle fusion rate constants and RRP-pool size by fitting a kinetic model to synaptic responses triggered by hypertonicity-induced SV fusion. Using this approach, we show that independent osmotic, genetic, and biochemical perturbations modulate SV release in a multiplicative/supralinear manner. The fact that linear (additive) effects on the energy barrier (activation energy) produce supralinear (multiplicative) effects on the release rate, helps to explain previously unexplained effects of genetic/pharmacological perturbations on synaptic transmission and provides a novel interpretation of the previously identified cooperative nature of $Ca^{2+}$-dependent release.

## Results

### Supralinear modulation of synaptic transmission by additive effects on the activation energy for vesicle fusion

Fusion of the lipid bilayer of synaptic vesicles with the plasma membrane involves deformation of membranes, dehydration of lipid head groups, neutralization of opposite membrane charges, and lipid splaying (*Kuzmin et al., 2001*; *Kozlovsky and Kozlov, 2002*; *Markin and Albanesi, 2002*; *Sorensen, 2009*), which together requires substantial energy. Vesicle priming and fusion can be represented in terms of an energy landscape, with energy barriers separating non-primed, primed, and fused states (*Figure 1A*) (*Sorensen, 2009*; *Walter et al., 2013*). The Arrhenius equation predicts an exponential relation between the rate constants of transitions between these states and the activation energies for these reactions, which correspond to the relative heights of these energy barriers (*Figure 1B*) (*Arrhenius, 1889a*, *1889b*; *Jahn and Grubmuller, 2002*). Hence, for transition from the primed to the fused state, the vesicle fusion-rate constant is given by

$$k_2 = Ae^{-\frac{E_a}{RT}},\qquad(1)$$

with $T$ the absolute temperature, $\bar{R}$ the gas constant, and $E_a$ the activation energy for synaptic vesicle fusion (*Figure 1A*). Since the speed of the reaction is determined by $E_a$ and not by the absolute height of the energy barrier for fusion, we use $E_a$ in the rest of this paper to explain effects on release kinetics. The prefactor $A$ is an empirical prefactor that takes into account the probability of collisions between reactants. For reactions in which the activation energy is low, this factor can limit release rates (diffusion limited reactions). Since SV fusion from the RRP proceeds from primed states where reactants are already positioned in close proximity and since fusion involves high-energy intermediate states, we assume that SV-release rates are predominantly governed by the activation energy and not by the number of collisions. Hence, we assume that changes in release rates most likely reflect changes in $E_a$ with constant $A$. In that case, if the activation energy for fusion at rest $E_{a,0}$ is reduced by an amount $\Delta E_1$ (*Figure 1C*), the corresponding new release rate constant is given by

$$
\begin{aligned}
k_{2,new} &= Ae^{-\frac{\left(E_{a,0}-\Delta E_1\right)}{RT}} \\
&= Ae^{-\frac{E_{a,0}}{RT}} e^{\frac{\Delta E_1}{RT}} \\
&= k_{2,0} \cdot m_1,
\end{aligned}
\qquad(2)
$$

with $m_1 = e^{\frac{\Delta E_1}{RT}}$ a multiplication factor and $k_{2,0} = Ae^{-\frac{E_{a,0}}{RT}}$ the rate constant for the $Ca^{2+}$-independent part of spontaneous release (*Xu et al., 2009*; *Ermolyuk et al., 2013*). Similarly, a further reduction of the activation energy with an amount $\Delta E_2$ by a second (independent) process (*Figure 1D*) leads to multiplication of the fusion-rate constant with an additional multiplication factor $m_2 = e^{\frac{\Delta E_2}{RT}}$,

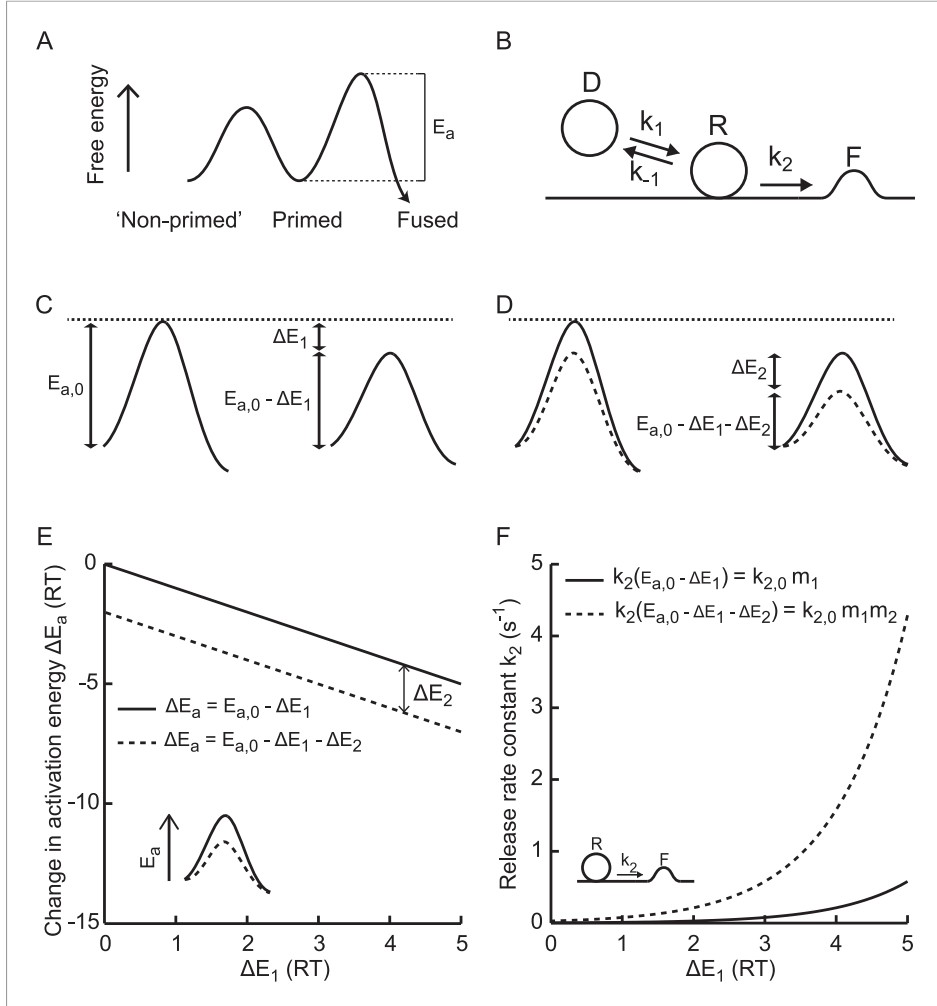

**Figure 1**. Supralinear modulation of synaptic efficacy through additive effects on the activation energy for fusion. (**A**) Schematic of the energy landscape for synaptic vesicle priming and fusion, with $E_a$ the activation energy for vesicle fusion, and (**B**) the corresponding vesicle-state model. (**C**) Reduction of the fusion activation energy at rest $E_{a,0}$ by an amount $\Delta E_1$, or (**D**) by a combined effect of $\Delta E_1$ and $\Delta E_2$. (**E**) Additive effect of $\Delta E_2$ causes a constant shift of the effective activation energy for fusion $\Delta E_a$ for different values of $\Delta E_1$, but a (**F**) multiplicative effect on the release rate constant $k_2$.

$$k_{2,new} = Ae^{-\frac{\left(E_{a,0}-\Delta E_1-\Delta E_2\right)}{RT}}$$
$$= k_{2,0} \cdot m_1 \cdot m_2. \tag{3}$$

This generalizes to

$$k_{2,new} = Ae^{-\frac{\left(E_{a,0}-\sum\limits_{i=1}^{N}\Delta E_i\right)}{RT}}$$
$$= k_{2,0} \cdot \prod_{i=1}^{N} m_i, \tag{4}$$

for $N$ independent reductions $\Delta E_i$ ($-\Delta E_i$ for enhancements) of the activation energy with corresponding multiplication factors $m_i = e^{\frac{\Delta E_i}{RT}}$. *Equation (4)* implies that additive effects on the activation energy for SV fusion result in multiplicative effects on the fusion rate (*Figure 1E,F*), which renders it a powerful way to modulate synaptic strength. In comparison, additive effects on the number of readily releasable vesicles cause additive effects on the fusion rate. We developed a method to quantify fusion rate constants from synaptic responses to hypertonic stimulation and tested whether osmotic, genetic, and biochemical perturbations modulate synaptic vesicle fusion rate in a supralinear manner.

## Minimal vesicle state model for synaptic vesicle release

Exposing neurons to hypertonic solution induces vesicle fusion selectively from the readily releasable pool (primed state) (*Rosenmund and Stevens, 1996*). This occurs by a mechanism that is not mediated by $Ca^{2+}$, as hypertonic sucrose (HS)-induced excitatory postsynaptic currents (EPSCs) are not changed when intracellular $Ca^{2+}$ is buffered by BAPTA, or when $Ca^{2+}$ influx through voltage gated $Ca^{2+}$ channels is blocked by $CdCl_2$ (*Rosenmund and Stevens, 1996*). HS-induced EPSCs display concentration-dependent changes in release kinetics, with higher degrees of hypertonicity leading to faster release, causing a decrease in time-to-peak and an increase in peak release rate (*Basu et al., 2007*) (*Figure 2A*). We applied a minimal vesicle state model, similar to *Weis et al. (1999)* (*Figure 1B*), and extended this with a time dependent description of the sucrose action on the release rate constant (*Figure 2B*, see 'Materials and methods' for mathematical description) to describe these release kinetics at various sucrose concentrations. EPSCs were simulated by modelling sucrose induced SV release rates and convolving them with a canonical miniature EPSC (see 'Materials and methods'). We found that—by varying only the maximal fusion rate constant $k_{2,max}$—our model reproduced all features in the experimental traces: a decrease in time-to-peak, an increase in peak release rate, and more release for increasing sucrose concentrations (*Figure 2B–C*) (*Figure 2—source data 1*). Above a given stimulus strength (0.5M sucrose in WT cells), the total amount of release remained constant, because the complete RRP was depleted, but peaks became larger and narrower when $k_{2,max}$ kept increasing. Latter features were also present in a reduced version of the model that neglects vesicle replenishment, which could be solved analytically (*Figure 2—figure supplement 1*). Hence, selective modulation of the fusion rate constant by HS stimulation in a simple vesicle state model is sufficient to describe characteristic features of synaptic responses to different levels of hypertonicity.

## Assessing RRP size and release rate constants

Next, we set out to fit HS-induced responses with our vesicle state model to assess synaptic release parameters including RRP, and rate constants for priming, unpriming, and fusion. Cultured autaptic neurons between DIV13-18 were challenged with HS concentrations ranging from 0.25–1M using a fast application system to establish a rapid transition from normal extracellular solution to hypertonic solution. In addition, spontaneous release was measured before cells were exposed to HS to quantify the release rate at 0M sucrose. The model accurately fitted synaptic responses induced by RRP depleting concentrations of 0.5M and higher, providing estimates for all model parameters (*Rosenmund and Stevens, 1996*; *Basu et al., 2007*) (*Figure 3A–C* and *Figure 3—figure supplement 1*). For 0.5M, we found a priming rate $k_1D$ of 0.132 ± 0.031 nC/s, which corresponded to 0.10 pool-units/s given an average pool size of 1.31 nC (see below) and was of the same order of magnitude as the 0.20 ± 0.03 pool-units/s at 25°C reported by *Pyott et al. (2002)*. The unpriming rate constant $k_{-1}$ at 0.5M was 0.11 ± 0.01 s$^{-1}$, corresponding to a RRP recovery time constant of $1/k_{-1} = 9.1$ s (see *Equation (21)*, 'Materials and methods'), which was of the same order of magnitude as recovery time constants reported in previous studies (10 s at 36°C (*Stevens and Tsujimoto, 1995*), 2.9 s at 32°C (*Toonen et al., 2006*), and 13 s (slow phase) at 25°C (*Pyott and Rosenmund, 2002*)). Priming and unpriming rates were not significantly different between different concentrations suggesting that these processes are not affected by hypertonic stimulation (*Figure 3—figure supplement 1*). We used estimations of the priming and unpriming parameters $k_1D$, and $k_{-1}$ to calculate RRP size from the steady state solution of the model given by *Equation (9)*, neglecting the value of $k_2$ before stimulation, which is three orders of magnitude smaller than $k_{-1}$ (compare *Figure 3C* and *Figure 3—figure supplement 1B*, *Figure 3—source data 1*). For stimulation with 0.5M, this yielded a RRP of 1.31 ± 0.23 nC,

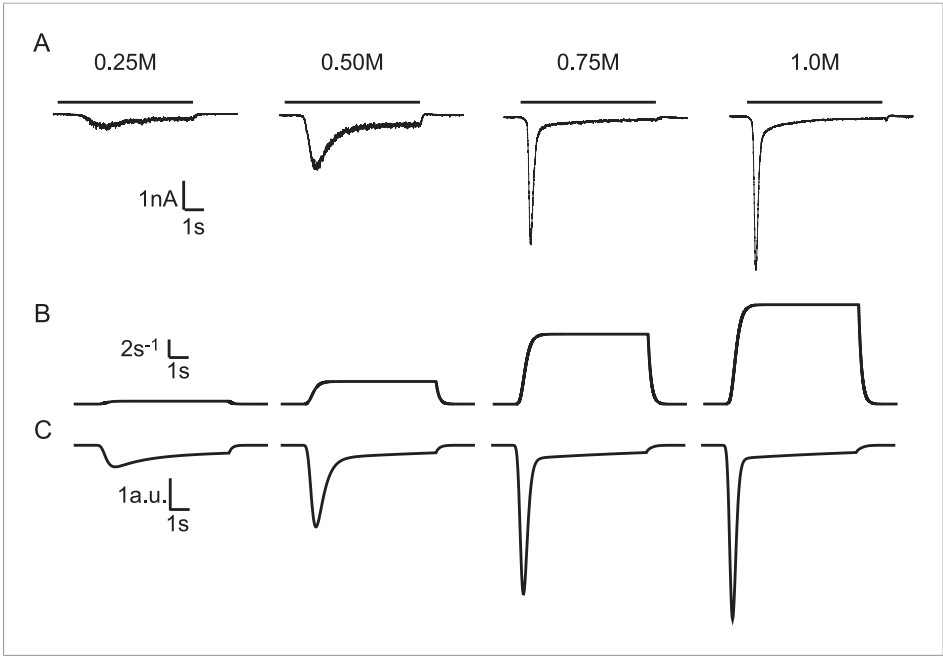

**Figure 2**. Modelling HS-induced EPSCs. (**A**) Concentration dependence of HS-induced release kinetics. (**B**) Model simulations of time courses of $k_2$, for different values of $k_{2,max}$ and (**C**) corresponding synaptic responses ($-k_2R$).

The following source data and figure supplement are available for figure 2:

**Source data 1**. Parameter values for *Figure 2—figure supplements 1 and, 3*.

**Figure supplement 1**. Analytical solution for hypertonic sucrose-induced release from a RRP without replenishment.

**Figure supplement 2**. Open tip experiments show rapid solution exchange.

**Figure supplement 3**. Effect of different model parameters on simulated HS-induced EPSCs.

corresponding to $11.9 \pm 2.4 \cdot 10^3$ (n = 12) vesicles, which was in the same range as reported for wild-type autaptic neurons by other studies ($15.9 \pm 2.9 \cdot 10^3$ (*Altrock et al., 2003*), $2.5 \pm 1.1 \cdot 10^3$ (*Augustin et al., 1999*), $5.36 \pm 0.87 \cdot 10^3$ (*Priller et al., 2006*), $24.7 \pm 5.6 \cdot 10^3$ (*Priller et al., 2007*), $17.2 \pm 3.0 \cdot 10^3$ (*Priller et al., 2009*), $6.35 \pm 0.9 \cdot 10^3$ (*Reim et al., 2001*), $11.0 \pm 1.2 \cdot 10^3$ (*Rhee et al., 2002*)). RRP sizes were similar for the RRP depleting concentrations of 0.5M and higher (*Figure 3B*). Our fit method yielded a more accurate estimate of the RRP size compared to quantification methods that use the charge transfer during the peak of the sucrose response and need to correct for on-going vesicle replenishment, either by subtracting the steady state current at the end of the response as a baseline (*Basu et al., 2007*; *Arancillo et al., 2013*) (*Figure 3—figure supplement 2A*), or by integrating the current to an arbitrary time-point after the peak (*Reim et al., 2001*; *Rosenmund et al., 2002*; *Toonen et al., 2006*; *Ikeda and Bekkers, 2009*) (*Figure 3—figure supplement 2B*). In addition, the rate constant for vesicle replenishment $k_1$ is one of the fitted model parameters, which allows the reconstruction of vesicle recruitment during sucrose application (see 'Materials and methods' and *Figure 3—figure supplement 2C*). We noticed that responses to 1M sucrose tended to have lower noise levels (*Figure 3A1*), which might point to an effect of receptor saturation and/or desensitization that was shown to be absent at 0.5M (*Pyott and Rosenmund, 2002*) but might play a role at higher concentrations. We confirmed that kinetics of responses to 0.5M were identical in the absence or presence of competitive AMPA receptor antagonist kynurenic acid (KYN), but found faster kinetics of 0.75M responses in the presence of KYN, suggesting that quantifications of model parameters obtained for concentrations higher than 0.5M should be interpreted with caution (*Figure 3—figure supplement 3*).

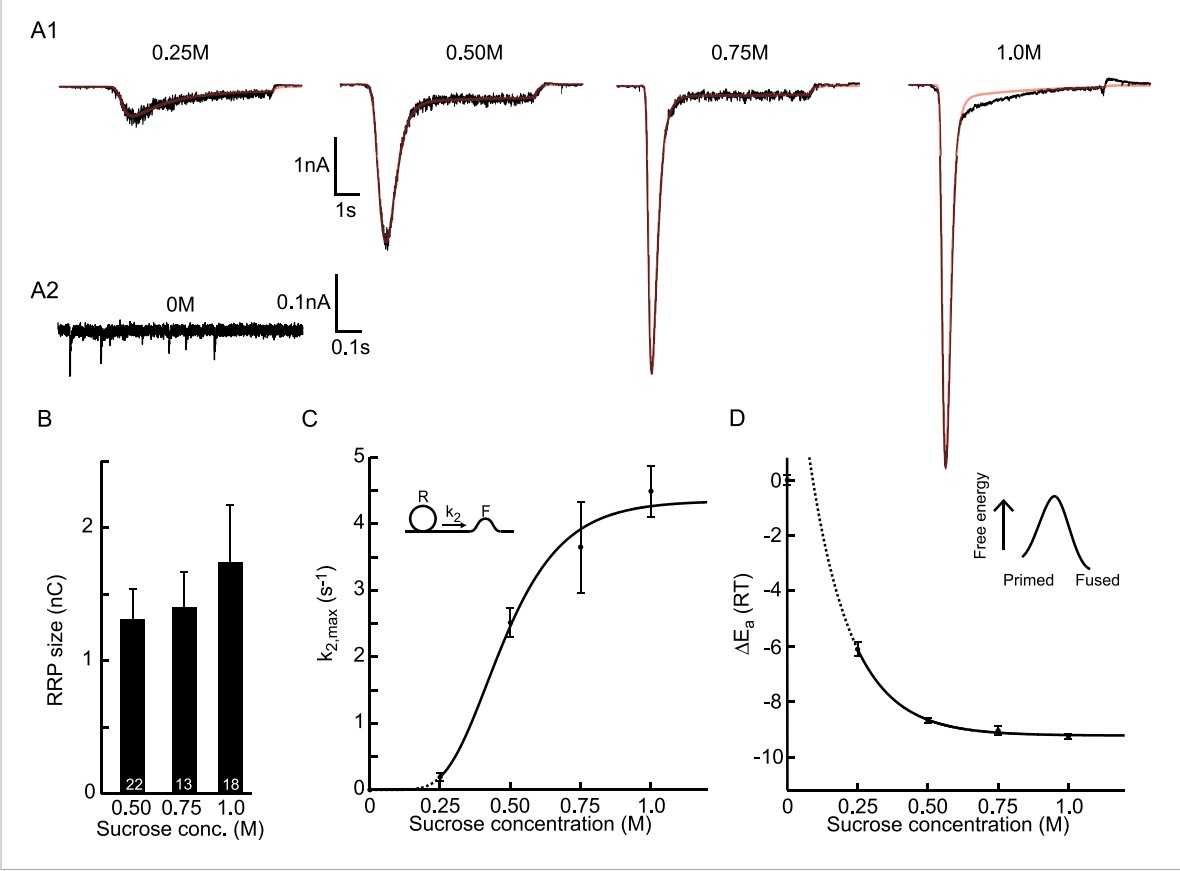

**Figure 3**. Probing the energy barrier for synaptic vesicle fusion. (**A1**) HS induced EPSCs (black) with model fits (red) superimposed. (**A2**) Spontaneous vesicle release at 0M sucrose. (**B**) RRP size obtained from model fits using *Equation (9)*. (**C**) Fitted maximal release rate constants $k_{2,max}$ at different sucrose concentrations. (**D**) Changes in activation energy (at 293 K) obtained from values for $k_{2,max}$ in **C** using *Equation (5)*. Data for 0.25M and higher were fitted with a monoexponential function, which was transformed into the dose–response curve in **C** using the equations given in *Figure 3—source data 1*.

The following source data and figure supplement are available for figure 3:

**Source data 1**. Parameter values for *Figure 3B–D*, bootstrap analysis *Figure 3*, *Figure 3—figure supplement 1A–C*, bootstrap analysis *Figure 3—figure supplement 1*, *Figure 3—figure supplement 3B–E*, and *Figure 3—figure supplement 4B–E*.

**Source data 2**. Parameter values for *Figure 3—figure supplement 5A and ,C*.

**Figure supplement 1**. Higher concentrations of hypertonic do not significantly affect upstream parameters but reduce the delay of sucrose action onset with respect to time of switching of the application barrel.

**Figure supplement 2**. Different methods to estimate RRP size from HS responses.

**Figure supplement 3**. Effect of the non-selective glutamate receptor antagonist kynurenic acid (KYN) on release kinetics.

**Figure supplement 4**. Subtraction of non-receptor current does not affect fitted model parameters.

**Figure supplement 5**. Fitting HS-induced EPSCs.

Maximal release rate constants $k_{2,max}$ were obtained from fits of responses to 0.25–1M sucrose. For non-depleting hypertonic stimulation (e.g., 0.25M), $k_{2,max}$ can be overestimated due to an underestimate of the RRP. Therefore, we fitted such current responses simultaneously with the response to a maximal depleting stimulation (e.g., 0.5M) from the same cell, keeping all the model

parameters the same between two stimulations, except $k_{2,max}$, $t_{del}$, and $\tau$. The release rate constant at 0M was obtained by dividing the frequency of spontaneously released events (mEPSCs) by the number of vesicles in the RRP (calculated by dividing the total RRP charge by the average mEPSC charge). However, this was probably an overestimation since the majority (>95%) of spontaneous release is Ca$^{2+}$-dependent, and intracellular Ca$^{2+}$ was not buffered in these experiments (*Xu et al., 2009*; *Groffen et al., 2010*). Ca$^{2+}$-dependent mEPSCs are most likely triggered by rapid spontaneous Ca$^{2+}$ fluctuations (SCFs) in the synaptic terminals, either caused by stochastic opening of voltage gated Ca$^{2+}$ channels ($\sim$50%) (*Goswami et al., 2012*; *Ermolyuk et al., 2013*) or release from intracellular calcium stores ($\sim$50%) (*Emptage et al., 2001*). This suggests that the frequency of these SCFs contributes with a constant $k_{2,SCFs}$ ($\sim$2–4 10$^{-4}$ s$^{-1}$) to the calculated release rate constant $k_{2,max}$, which dominates at 0M sucrose but is negligible compared to fusion rate constants induced with higher concentrations (*Figure 3—source data 1*). In contrast to the other fitted model parameters, $k_{2,max}$ was significantly different between different concentrations and showed a sigmoidal dependence on sucrose concentration (*Figure 3C*). The values for $k_{2,max}$ at 0.75 and 1M might be underestimated due to receptor saturation as discussed above (*Figure 3—figure supplement 3*).

## Sucrose stimulation reflects a decrease in the activation energy for fusion

As we argued above, Ca$^{2+}$-triggered exocytosis belongs to a class of reactions that are likely to be limited by activation energy, rather than by the frequency of collisions between reactants. This follows from the preassembly of a fusion machinery during vesicle priming, and from the expected existence of high-energy intermediates. During stimulation with hypertonic solution, drawing water from the cell will increase the concentration of reactants. This might increase collision rates proportional with the increased concentration, but this is unlikely to account for the 10$^4$-fold increase in $k_{2,max}$. Moreover, the (moderate) increase in reactant concentration might be counteracted by molecular crowding effects and increases in viscosity (*Miermont et al., 2013*). Consistent with this notion, we observed that upstream steps in the exocytotic cascade, which are in fact more likely to be collision limited (such as vesicle docking and priming, reflected in the overall priming rate $k_1D$), showed a tendency to *decrease* with high osmolarity (*Figure 3—figure supplement 1*), indicating that molecular crowding/viscosity dominates the effect of increased reactant concentration. Overall, we conclude that a HS challenge is most likely to change fusion through a change of the activation energy for fusion (i.e., the exponential factor in the Arrhenius equation), rather than the pre-exponential factor $A$.

Changes in activation energy for fusion follow from changes in $k_{2,max}$ using *Equation (1)* assuming $A$ is constant,

$$\begin{aligned}
\Delta E_a \quad &= E_{a,1} - E_{a,2} \\
&= \bar{R}T\left(\ln(A) - \ln\left(k_{2,max,1}\right)\right) - \bar{R}T\left(\ln(A) - \ln\left(k_{2,max,2}\right)\right) \\
&= \bar{R}T\left(\ln\left(k_{2,max,2}\right) - \ln\left(k_{2,max,1}\right)\right).
\end{aligned} \tag{5}$$

*Figure 3D* depicts the calculated changes in activation energies corresponding to the changes in $k_{2,max}$ for different sucrose concentrations in *Figure 3C*. We find that the maximal reduction in the activation energy for fusion by 1M sucrose is $9.3\bar{R}T$. This value is probably about $3\bar{R}T$ too low since (as discussed above) $k_{2,max}$ is overestimated at 0M (up to 20 fold), but not at higher sucrose concentrations. Expressed in units of kCal/mol, the HS-induced change in activation energy corresponds to 5.4 kCal/mol, which is comparable to the estimated reduction of 5.9 kCal/mol during the action potential (*Rhee et al., 2005*). Hence, fusion rate constants obtained from fitting HS-induced synaptic responses to a minimal vesicle-state model can be used to calculate changes in activation energy for fusion, which enables to study this parameter under different experimental conditions.

## Relationship between release kinetics and RRP depletion

The extent of RRP depletion upon application of submaximal sucrose has been used as a measure of 'release willingness' or 'fusiogenicity' of vesicles, which is proposed to be inversely related to the energy barrier for fusion (*Basu et al., 2007*; *Gerber et al., 2008*; *Xue et al., 2010*; *Rost et al., 2011*). To investigate whether changes in the activation energy for fusion can explain changes in the depleted RRP fraction at submaximal sucrose, we analyzed the relation between release kinetics ($k_{2,max}$) and RRP depletion in the model and compared this with experimental data. The depleted RRP fraction was defined as the fraction of the RRP depleted by a submaximal HS stimulus relative to

a maximal depleting stimulus (0.5M sucrose). Simulations applying 7 s HS-stimulations for different values of $k_{2,max}$ yielded a linear relation for low values of $k_{2,max}$, which levels off and saturates to 1 (complete depletion) at high $k_{2,max}$. This relation transforms into a sigmoidal curve when $k_{2,max}$ is plotted on a $\log_{10}$ scale (black line in *Figure 4B*) and can be approximated by an analytically derived function (see 'Materials and methods' and *Figure 4—figure supplement 1*) (*Figure 4—source data 1*). The value for $k_{2,max}$, that we experimentally find with 0.5M stimulation, predicts only a 94% depletion of the RRP implying that up to 6% more release is expected with higher concentrations. However, in practice, these slightly larger responses might be difficult to detect because of receptor saturation and desensitization effects at these concentrations. We experimentally confirmed the predicted relation with data points from submaximal 0.25M responses being distributed along the steep phase of the curve (*Figure 4A,B*). As expected, 0.75 and 1M responses yielded high values for $k_{2,max}$ and complete RRP depletion. These results show that a change in $k_{2,max}$ only is sufficient to explain changes in the depleted RRP fraction: with slow release kinetics (low $k_{2,max}$), the RRP is not effectively depleted, because of on-going refilling (priming), whereas from a certain value of $k_{2,max}$ the amount of RRP depletion is maximal, but depletion occurs with faster kinetics. Hence, with this relation the extent of RRP depletion in response to different sucrose concentrations can be used to discriminate between effects on release kinetics and priming. Maximally depleting stimuli report the RRP, while changes in the depleted RRP-fraction at submaximal (e.g., 0.25M) stimuli are an indication of changes in $k_{2,max}$, indicative of changes in the activation energy for fusion.

## Modulation of the activation energy for fusion by genetic and biochemical perturbations

Next, we investigated the additivity between osmotic and genetic or biochemical perturbations on release kinetics and RRP depletion. We extracted data from literature on genetic and/or biochemical perturbations with an effect on the release willingness of vesicles. Interestingly, changes in release

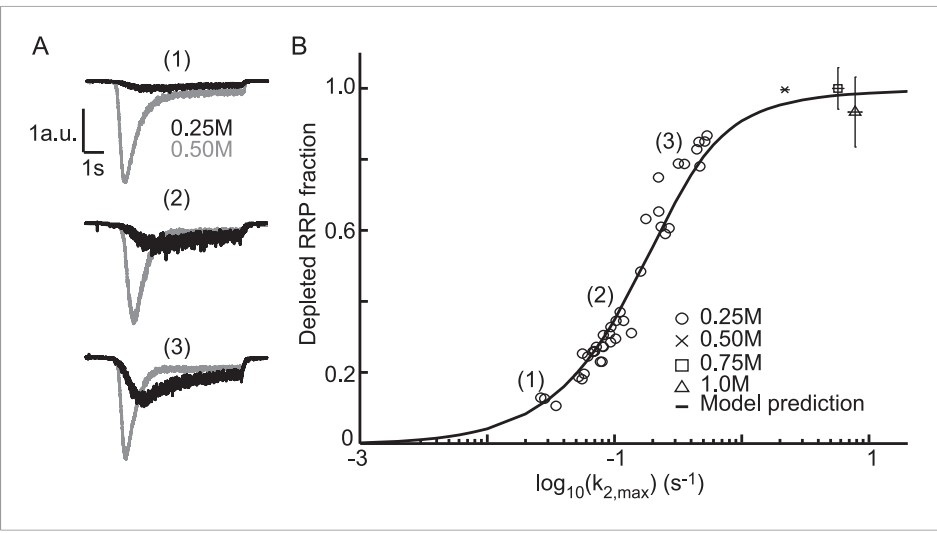

**Figure 4**. Relation between depleted RRP fraction and release kinetics. (**A**) Examples of submaximal responses in different cells. 0.25M responses (black), scaled to 0.5M responses (grey) in the same cell, display faster kinetics when a larger fraction of the RRP is depleted. (**B**) Fitted data overlayed on the predicted curve. Data points corresponding to the examples in **A** are indicated. Data points for 0.50M, 0.75M, and 1.0M are shown as mean ± SEM. Note that whereas the model predicts a 94% depletion of the RRP with 0.5M the y-axis value at 0.5M is one per definition since the RRP size at this concentration was used as a reference to calculate the depleted RRP fraction.

The following source data and figure supplement are available for figure 4:

**Source data 1**. Parameter values for *Figure 4B* and *Figure 4—figure supplement 1*.

**Figure supplement 1**. Comparison of analytical approximation and model predictions of the relation between release kinetics and RRP depletion.

willingness were reported for proteins with distinct presynaptic functions, including the priming factor Munc13, the tSNARE Syntaxin, the SNARE-complex binding protein Complexin, and the metabotropic GABA$_B$ receptor (*Basu et al., 2007*; *Gerber et al., 2008*; *Xue et al., 2010*; *Rost et al., 2011*). We retrieved for different types of perturbations, the reported depleted RRP fractions, and corresponding peak release rates, defined as the release rate at the peak of the HS-induced response (*Basu et al., 2007*). Plotting these data points in one graph showed the same non-linear relation between release kinetics and RRP depletion for the four different data sets (*Figure 5*). To compare this experimentally observed relation with our model prediction, we simulated sucrose responses for different values of $k_{2,max}$, keeping all other parameters constant, and calculated peak release rates and corresponding depleted RRP fractions from the simulated traces in the same way as was done for the experimental traces (*Figure 3—figure supplement 2A*) (*Figure 5—source data 1*). The model prediction of the relation between release kinetics and RRP depletion was in good accordance with the experimental data (*Figure 5*). Hence, this non-linear dependence can be explained by changes in the release rate constant $k_{2,max}$ only.

## Supralinear modulation of release kinetics by phorbol esters and complexins through additive effects on the activation energy

Next, we tested whether these biochemical and genetic perturbations modulate release kinetics in a supralinear manner, measuring release rate constants at different sucrose concentrations between 0 and 0.5M to avoid effects of receptor saturation and desensitization. Phorbol ester is known to potentiate synaptic release in a number of systems (*Searl and Silinsky, 1998*; *Rhee et al., 2002*; *Basu et al., 2007*; *Wierda et al., 2007*; *Lou et al., 2008*). First, we recorded spontaneous release and

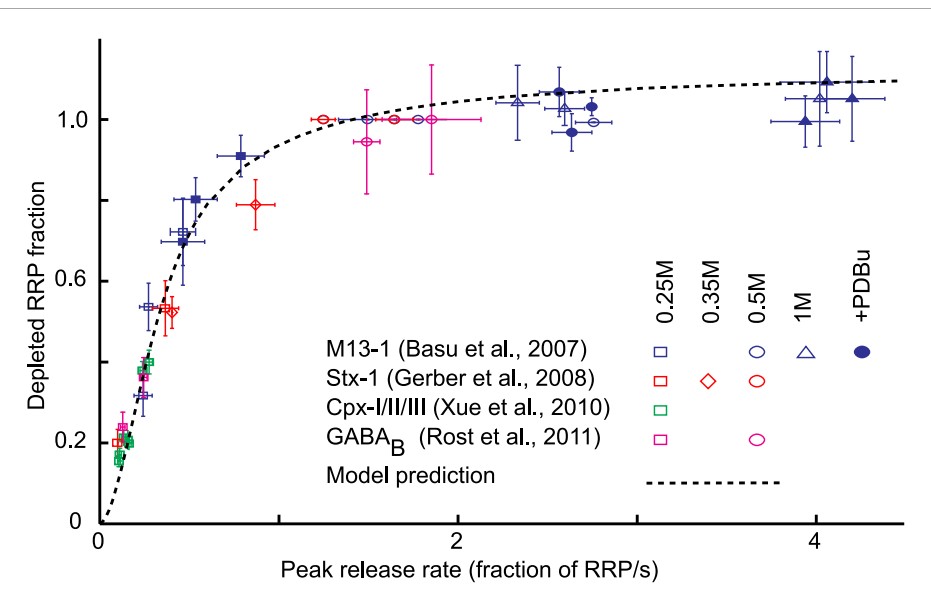

**Figure 5**. Model predicts relation between peak release rate, defined as the release rate at the peak of a HS-induced response, and depleted RRP fraction for different combinations of HS stimulations and genetic or biochemical manipulations of the activation energy for fusion. Data are taken from (*Basu et al., 2007*; *Gerber et al., 2008*; *Xue et al., 2010*; *Rost et al., 2011*) Model prediction is obtained from peak release rates and depleted RRP fractions extracted from model simulations where parameter $k_{2,max}$ is varied keeping other model parameters constant. Note that beyond 0.5M the predicted curve and some data points overshoot the value of one because 0.5M was used as a reference to calculate the depleted RRP fraction at the other concentrations, assuming complete depletion at 0.5M, whereas the model predicts only 94% depletion at this point.

The following source data is available for figure 5:

**Source data 1**. Parameter values for *Figure 5*.

responses to 0.2–0.5M hypertonic stimulations, before and after PDBu application (1 µM) (*Figure 6—figure supplements 1, 2*). We observed potentiation of the spontaneous release and submaximal (0.2–0.4M) responses as well as faster kinetics for the 0.5M response, but no effect on RRP size or priming and unpriming rate constants (*Figure 6A*, *Figure 6—figure supplement 3*). When comparing the effect of PDBu on release kinetics between different sucrose concentrations, indeed a supralinear increase in $k_{2,max}$ was found, with the increase in $k_{2,max}$ being three orders of magnitude larger at 0.5M than at 0M (*Figure 6B*, *Figure 6—source data 1*). Next, we calculated the activation energies from the changes in $k_{2,max}$, using *Equation (5)*, which were reduced with a similar $\Delta E_a$ for all sucrose concentrations (*Figure 6C*, *Figure 6—source data 1*). This multiplicative effect on release kinetics, but additive effect in the activation energy domain, became more evident when absolute changes in these variables were plotted, with an exponential increase in $k_{2,max}$ and a $\sim -0.3$ $\bar{R}T$ shift in the fusion-activation energy for 0.2–0.5M sucrose (*Figure 6D–E*). The almost twofold higher decrease at 0M was probably an overestimation because of the increased sensitivity to spontaneous $Ca^{2+}$ fluctuations after PDBu, which will increase the contribution of $k_{2,SCFs}$ to $k_{2,max}$, again dominating $k_{2,max}$ at 0M but being negligible at higher concentrations.

Next, we reanalysed the raw responses to 0, 0.25, and 0.5M sucrose in complexinI/II deficient neurons and their controls from a study by *Xue et al. (2010)*. Whereas responses to 0.5M did not differ in released RRP size, and priming and unpriming were not affected (*Figure 7A*, *Figure 7—figure supplement 1*), a markedly reduced fraction of the RRP was released by 0.25M stimuli in the null mutants, suggesting an increased activation energy for fusion in the absence of complexins. Indeed, release kinetics were slowed down as predicted by the relation between $k_{2,max}$ and depleted RRP fraction (*Figure 7A*, *Figure 7—figure supplement 1D*). This effect of complexin deletion on release kinetics was supralinear with an eightfold larger reduction of $k_{2,max}$ at 0.5M than at 0.25M, whereas the corresponding activation energies shifted with 0.4 and 0.8 $\bar{R}T$ at these concentrations (*Figure 7B–E*). The overall supralinearity is in line with an activating role of complexin in exocytosis by a reduction of the activation energy for fusion (*Figure 7B–C*, *Figure 7—source data 1*). However, the reduction of the activation energy was less at 0M, and also seemed less at 0.5M than at 0.25M (*Figure 7E*), possibly indicating that complexins exert several effects, for instance clamping a secondary $Ca^{2+}$ sensor for spontaneous and asynchronous release, rendering the synapse more sensitive to spontaneous $Ca^{2+}$ fluctuations (*Yang et al., 2010*; *Ermolyuk et al., 2013*). Another possibility is that complexin also affects the frequency factor, for example, because the absence of complexin changes the cooperativity of exocytosis.

## Discussion

We developed a vesicle state model that can accurately reproduce synaptic responses to varying hypertonicity of both published data and new experiments reported here. This model can be exploited to obtain accurate estimates of the RRP, priming-, unpriming-, and fusion rate constants. It shows that independent osmotic, biochemical, and genetic perturbations produce supra-linear modulatory effects on the fusion rate.

### Kinetic analysis provides essential release parameters from a $Ca^{2+}$-independent stimulus

Exploiting the kinetic model presented here to assess essential release parameters like RRP-size and fusion kinetics from HS-induced responses has advantages over existing methods. Firstly, this model uses the steady state solution (*Equation (9)*) to calculate the RRP size. This circumvents the necessity to correct post-hoc for RRP replenishment during the stimulus as in other RRP estimation methods (*Schneggenburger et al., 1999*; *Moulder and Mennerick, 2005*) (*Figure 3—figure supplement 2A,B*). Secondly, the relation between release kinetics and RRP depletion can be used to predict changes in $k_{2,max}$ from changes in the depleted RRP fraction. This makes it possible to discriminate between changes in the activation energy (indicated by changes in the depleted RRP fraction tested with submaximal HS stimuli (*Xue et al., 2010*; *Arancillo et al., 2013*)) and priming effects (indicated by changes in the response to maximal depleting HS stimuli). An important consequence is that in situations where the activation energy is increased (e.g., by genetic deletion of a gene that reduces the energy barrier for fusion), 0.5M sucrose might not be enough to fully deplete the RRP. This could be erroneously interpreted as a priming defect. Thirdly, our model also quantifies priming- and unpriming-rate constants

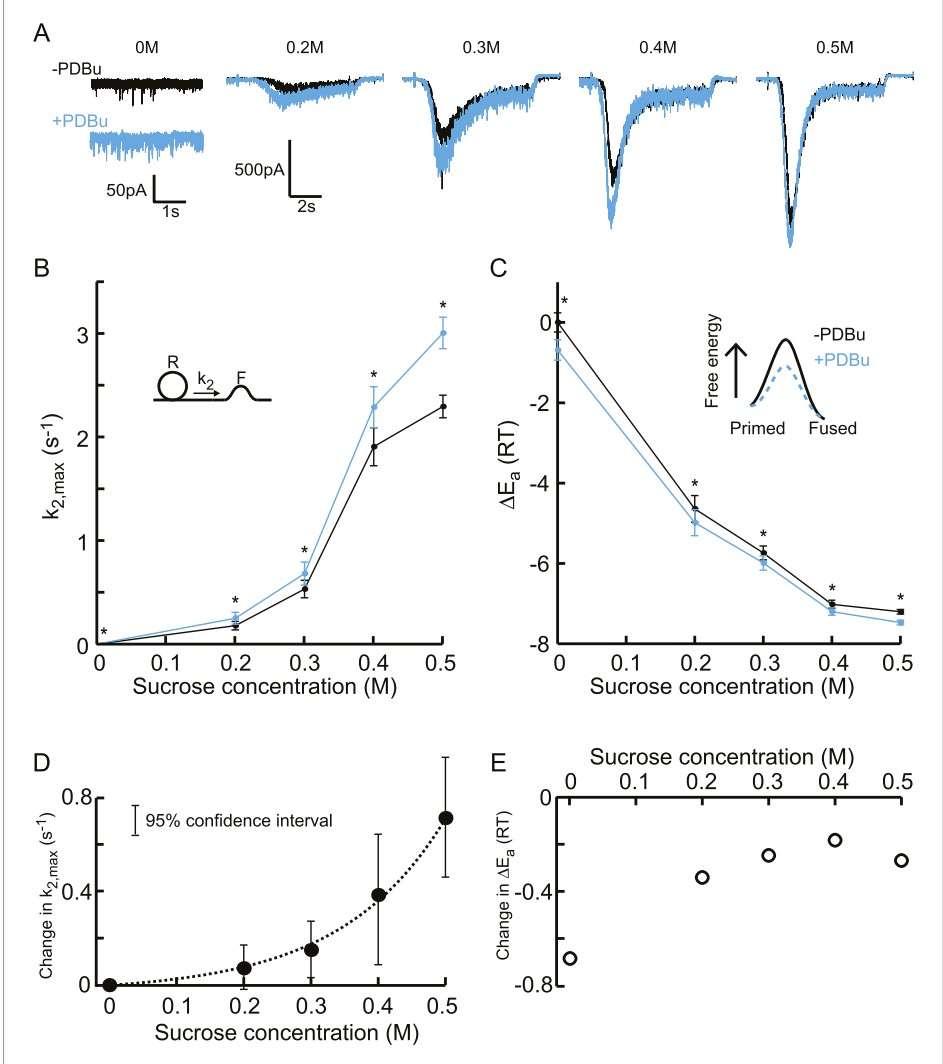

**Figure 6**. Additive effect on the activation energy for fusion induced by PDBu causes supralinear effect on release kinetics. (**A**) Current traces, (**B**) release rate constants $k_{2,max}$, and (**C**) activation energies for fusion at different sucrose concentrations in the absence and presence of PDBu. PDBu-induced changes in $k_{2,max}$ and $\Delta E_a$, obtained by subtraction of the data curves in **B** and **C** before and after PDBu application, show (**D**) an exponential increase in $k_{2,max}$ for increasing sucrose concentrations whereas (**E**) the changes in the energy domain are in the same order of magnitude (reduction at 0M is probably an overestimation due to $Ca^{2+}$ depenence of the spontaneous release, [see text]). Mean values of $k_{2,max}$ displayed are all within the 95% confidence interval as determined by Bootstrap analysis.

The following source data and figure supplement are available for figure 6:

**Source data 1**. Parameter values for *Figure 6B–E*, bootstrap analysis *Figure 6*, *Figure 6—figure supplement 3A–D*, and *Figure 6—figure supplement 3*.

**Figure supplement 1**. Random examples of individual HS-evoked EPSCs (black) in the absence of PDBu, overlaid with their best fit (red).

**Figure supplement 2**. Random examples of individual HS-evoked EPSCs (blue) in the presence of PDBu, overlaid with their best fit (red).

**Figure supplement 3**. Upstream parameters and RRP size are not affected by PDBu application.

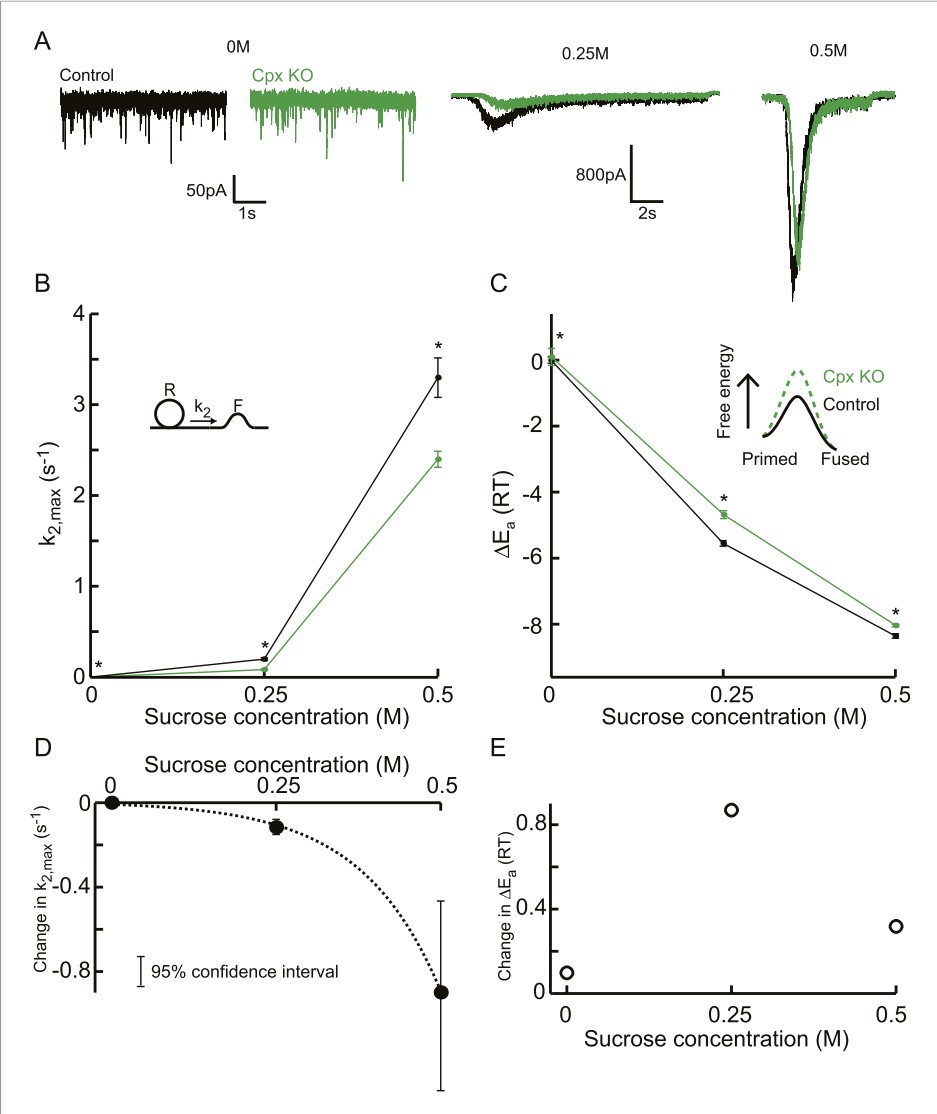

**Figure 7.** Additive effect on the activation energy for fusion induced by Cpx deletion causes supralinear effect on release kinetics. (**A**) Current traces, (**B**) release rate constants $k_{2,max}$, and (**C**) fusion energy barrier heights at different sucrose concentrations for control and CpxKO cells. Cpx deletion-induced changes in $k_{2,max}$ and $\Delta E_a$, obtained by subtraction of the data curves for control and CpxKO in **B** and **C**, show (**D**) an exponential increase in $k_{2,max}$ for increasing sucrose concentrations whereas (**E**) the changes in the energy domain are in the same order of magnitude. Mean values of $k_{2,max}$ displayed are all within the 95% confidence interval as determined by Bootstrap analysis. Cpx data were published before in (*Xue et al., 2010*) and reanalysed here.

The following source data and figure supplement are available for figure 7:

**Source data 1**. Parameter values for, bootstrap analysis *Figure 7B–E*, bootstrap analysis *Figure 7*, *Figure 7—figure supplement 1A–D*, and bootstrap analysis *Figure 7—figure supplement 1*.

**Figure supplement 1**. Upstream parameters and RRP size are not affected in Cpx KO.

($k_1$ and $k_{-1}$), which for instance allows reconstruction of the time course of replenishment during HS stimulation at resting $Ca^{2+}$ levels. Finally, all model parameters mentioned above are quantified using a $Ca^{2+}$-independent stimulus, which to a large extent excludes differences in $Ca^{2+}$ signalling or $Ca^{2+}$ sensitivity as confounding factors.

## The Arrhenius equation infers the activation energy for synaptic vesicle fusion

Since activation energies cannot be directly measured in synapses, we used the Arrhenius equation to infer these from HS-induced release rate constants. Four arguments suggest that the effect of hypertonic solution (HS) on synaptic release is primarily due to a reduction in activation energy, and not by an increase in the number of collisions as a result of shrinkage (accounted for by the Arrhenius pre-exponential factor A). First, exocytosis is expected to take place via a sequence of high-energy intermediates, together determining the activation energy for fusion (see 'Discussion' below). Therefore, modulation of the fusion activation energy is a plausible efficient route to regulate vesicle fusion. Second, HS specifically releases primed vesicles (*Rosenmund and Stevens, 1996*), which are bound to the plasma membrane with the fusion machinery preassembled. Thus, fusion is unlikely to be diffusion limited. Third, rapid cell shrinking can have opposite effects on the number of collisions, which are expected to affect priming/unpriming and fusion rates similarly. It can either increase the collision frequency due to an increase in the concentrations of reactants or (given the already high protein concentrations in synapses (*Wilhelm et al., 2014*)) decrease collision frequency because of molecular crowding and viscocity effects (*Miermont et al., 2013*). Since upstream docking/priming steps displayed a trend towards a *decrease* upon higher HS application, molecular crowding seems to offset any effect on reactant concentration, and therefore, the drastic increase in fusion rate cannot be attributed to A via an increased collision rate. Finally, the reduction in activation energy identified here (6.1 $\bar{R}T$ for 0.25M) (*Figure 3D*) is comparable to the reduction expected by HS stimulation (0.2M) of liposome fusion on theoretical grounds (~7 $\bar{R}T$ (*Malinin and Lentz, 2004*)). Nevertheless, manipulations that change the pre-exponential factor will also contribute to changes in the fusion rate of vesicles in the presence of HS.

## Activation energy modulation is a powerful way to regulate synaptic transmission

Many factors influence synaptic release probability, such as RRP size, modulation of $Ca^{2+}$-and $K^+$-channel properties, $Ca^{2+}$-buffering/diffusion, and the sensitivity of $Ca^{2+}$ sensors (*Neher and Sakaba, 2008*; *Fioravante and Regehr, 2011*). Changes in the activation energy are suggested to affect release probability by rendering vesicles more/less fusogenic (*Basu et al., 2007*; *Wierda et al., 2007*; *Gerber et al., 2008*; *Xue et al., 2010*). This is a powerful way to regulate synaptic transmission because of its exponential effect on the fusion rate, whereas RRP size modulation affects synaptic transmission in a proportional fashion (*Sakaba and Neher, 2001*; *Rhee et al., 2002*; *Lipstein et al., 2013*; *Walter et al., 2013*). A well-studied example is the facilitatory effect of diacylglycerol (DAG) analogues such as phorbol esters on AP induced release. DAG activates two interdependent pathways: direct activation of Munc13 via its $C_1$ domain and PKC dependent phosphorylation of Munc18. Together, these events reduce the energy barrier for fusion, potentiate vesicular release probability after high frequency stimulation, and produce faster synaptic depression (*Rhee et al., 2002*; *Basu et al., 2007*; *Wierda et al., 2007*; *Garcia-Perez and Wesseling, 2008*; *de Jong and Verhage, 2009*; *Genc et al., 2014*). Other presynaptic proteins may also contribute to activation energy reductions (*Gerber et al., 2008*; *Weber et al., 2010*; *Xue et al., 2010*; *Rost et al., 2011*). This suggests that there are either multiple ways by which proteins can modulate the activation energy for fusion or that they all converge onto the same process (e.g., SNARE formation/stabilization) controlling the activation energy. Interestingly, a model of additive modulation of the activation energy implies that molecules can exert their effect independently and do not necessarily need to interact physically to produce complex supra-linear effects on synaptic transmission.

## Additive effects on the activation energy might explain $Ca^{2+}$ cooperativity of synaptic vesicle release

$Ca^{2+}$ controls vesicle fusion in a cooperative fashion (*Dodge and Rahamimoff, 1967*). This has been extensively studied in the Calyx of Held showing that a 3 orders of magnitude increase in $Ca^{2+}$ give rise to a 6 orders of magnitude increase in the vesicle fusion rate (*Schneggenburger and Neher, 2000*; *Lou et al., 2005*; *Neher and Sakaba, 2008*). This supra-linear relationship can be well described by a phenomenological model for 'allosteric' modulation of the presynaptic $Ca^{2+}$ sensor (*Lou et al., 2005*), which captures the low cooperativity (<1) for triggering vesicle fusion at basal $Ca^{2+}$

and high Ca²⁺ cooperativity (~4) at Ca²⁺ concentrations beyond 5 μM (*Figure 8A*). However, we note that the exact same model follows from *Equation (4)* when assuming that the Ca²⁺ sensor reduces the activation energy with an amount $\Delta E_{Ca}$ for each Ca²⁺-ion binding. In this model (as in the previous model (*Lou et al., 2005*)), a vesicle can be in one of six different states depending on how much Ca²⁺ ions are bound to the Ca²⁺ sensor associated with the vesicle. From each state, release will occur with a specific fusion rate constant

$$k_{2,n} = l_+ f^n, \tag{6}$$

with $l_+ = k_{2,0}$ the basal fusion rate constant, $f = e^{\frac{\Delta E_{Ca}}{RT}}$ a multiplication factor, and $n$ the number of Ca²⁺ ions bound to the Ca²⁺ sensor (*Figure 8B*). In line with our findings here, the fusion promoting effect of PDBu, described in Lou et al. by the increase of the spontaneous release rate constant $l_+$ (*Lou et al., 2005*), corresponds to a $\Delta E_{PDBu}$ reduction of the activation energy resulting in a new rate constant $l_{+,new} = l_+ e^{\frac{\Delta E_{PDBu}}{RT}}$.

All together, this suggests that the Ca²⁺ sensor modulates fusion supralinearly through additive effects on the fusion activation energy. As a consequence, other factors (such as PDBu) do not necessarily need to interact directly with the sensor to modulate the Ca²⁺ sensitivity of release, but can exert their effect on the activation energy independently.

## Multiple (independent) molecular events may underlie changes in the activation energy for fusion

Membrane fusion is a complex process assumed to proceed via a stalk intermediate, with many steps contributing to the activation energy for fusion (*Jahn and Grubmuller, 2002*; *Kozlovsky and Kozlov, 2002*). A state immediately preceding stalk formation may consist of 'splayed' lipids, which have left their native leaflet and form a high-energy intermediate (*Risselada and Grubmuller, 2012*). Formation and zippering of the SNARE-complex allows the membranes to approach closely (*Lindau et al., 2012*) and might also induce or support lipid splaying directly along the linker regions of syntaxin and synaptobrevin/VAMP (*Risselada et al., 2011*). Molecular changes in these proteins, changes in their number or stoichiometry, and/or association/dissociation of additional factors such as complexins,

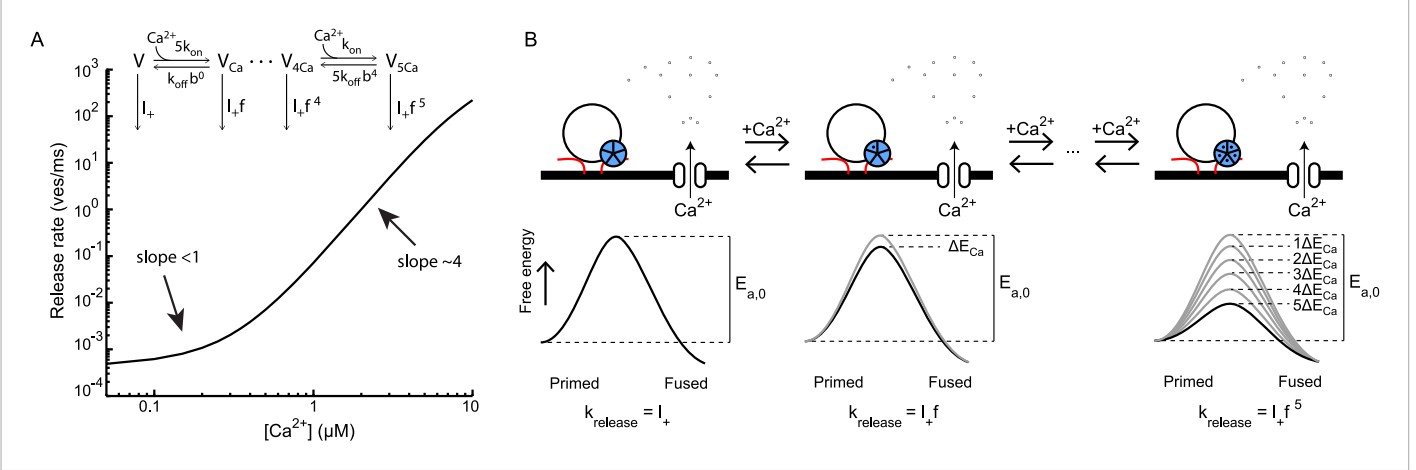

**Figure 8**. Supralinear Ca²⁺ dependency of release can be explained by additive modulation of the activation energy for fusion by the Ca²⁺ sensor. (**A**) Non-linear relation between Ca²⁺ and release rate in the Calyx of Held as predicted by the allosteric model of *Lou et al. (2005)*. Allosteric model with 6 different vesicle states ($V$, $V_{Ca}$, · · ·, $V_{5Ca}$) is depicted in inset. (**B**) Reinterpretation of this allosteric model in terms of additive effects on the activation energy of the binding of Ca²⁺ to the Ca²⁺ sensor: each Ca²⁺ ion that binds reduces the activation energy $E_{a,0}$ by an amount $\Delta E_{Ca}$. From *Equation (4)* it follows that for each vesicle state the release rate constant krelease is given by *Equation (6)*, with $l_+ = A e^{-\frac{E_{a,0}}{RT}}$ the spontaneous release rate constant and $f = e^{\frac{\Delta E_{Ca}}{RT}}$ a multiplication factor. This is mathematically equivalent to the release rate constants depicted for the different vesicle states in the allosteric model in **A** and thus yields the same prediction of the non-linear relation between Ca²⁺ and release rate.

Munc13, or Munc18 may all lower the activation energy (*Gerber et al., 2008*; *Li et al., 2011*; *Ma et al., 2011*).

Whether or not SNARE-complexes are already (partly) assembled at the time when APs open $Ca^{2+}$ channels is a matter of intense debate (*Jahn and Fasshauer, 2012*). The energy released during the formation of a SNARE-complex has been estimated to range between 20 and 35 $\bar{R}T$ (*Mohrmann and Sorensen, 2012*), which is 2–3 times higher than what we find for 1M sucrose. However, in case, SNARE-complexes are partly preassembled, only part of the estimated energy would become available for fusion when HS would promote full assembly (see review (*Sorensen, 2009*)). Furthermore, the similar values of HS-induced reduction in activation energy, identified here and in a theoretical study of protein-free liposome fusion (*Malinin and Lentz, 2004*), indicate that the effect of hypertonicity might be on the lipids themselves, by helping to fill energetically expensive 'voids' that form during fusion (*Malinin and Lentz, 2004*). If this is the case, several other molecules might act in similar ways, including $Ca^{2+}$-bound synaptotagmin and SNAREs, and several accessory proteins that also interact directly with lipids (*Seiler et al., 2009*; *Shin et al., 2010*). The actions of a small number of accessory proteins like complexin, Munc13, CAPS, and Munc18, and the proposed stoichiometry of SNARE-complexes per vesicle (*Sinha et al., 2011*; *van den Bogaart et al., 2010*; *Mohrmann et al., 2010*) provide all the necessary input for molecular-dynamic models (*Lindau et al., 2012*) to resolve the exact nature of the synaptic vesicle fusion process. Kinetic analysis of HS induced synaptic responses will be highly instrumental to test predictions from such models.

## Materials and methods

### Electrophysiological recordings

Autaptic hippocampal neurons from wild-type mice were grown for 13–18 days on glia island cultures before measuring. Whole-cell voltage-clamp recordings (Vm = −70 mV) were performed at room temperature (20–24°C) with borosilicate glass pipettes (2.5–4.5 MOhm) filled with 125 mM $K^+$-gluconic acid, 10 mM NaCl, 4.6 mM $MgCl_2$, 4 mM $K_2$-ATP, 15 mM creatine phosphate, 10U/ml phosphocreatine kinase, and 1 mM EGTA (pH 7.30). External solution contained the following (in mM):10 HEPES, 10 glucose, 140 NaCl, 2.4 KCl, 4 $MgCl_2$, and 4 $CaCl_2$ (pH = 7.30, 300 mOsmol). Recordings were acquired with an Axopatch 200A amplifier (Molecular Devices, Sunnyvale CA), Digidata 1322A, and Clampex 9.0 software (Molecular Devices). After whole cell mode was established, only cells with a leak current of <250 pA were accepted for analysis. $Ca^{2+}$-independent vesicle release was evoked by hypertonic solutions consisting of external solution containing 0.25, 0.5, 0.75, or 1M sucrose. Gravity infused external solution was alternated with 7 s of perfusion with hypertonic solution by rapidly switching between barrels within a custom-made tubing system (FSS standard polyamine coated fused silica capillary tubing, ID 430 µm, OD550 µm, Postnova analytics, Landsberg am Lech, Germany) attached to a perfusion Fast-Step delivery system (SF-77B, Warner instruments corporation, Hamden CT) and directed at the neuron. Solution flow was controlled with an Exadrop precision flow rate regulator (B Braun, Melsungen, Germany) to assure all sucrose solutions flowed with a rate of 0.5 ml/min irrespective of differences in viscosity. Using this system, solution exchange was complete within 0.4 s as measured by the change in holding current after switching from normal (0.3M) to 10 times diluted (0.03M) extracellular solution containing 0.5 or 1M sucrose in an open-tip experiment (*Figure 2—figure supplement 2*). Therefore, solution exchange can be considered instantaneous compared to the induced postsynaptic currents, which respond with a delay of 1.1 (1M)–1.6 s (0.25M) (*Figure 3—figure supplement 1C*). Multiple sucrose solutions with various concentrations were applied to the same cell, taking a 1–2 min rest period in between solutions to accommodate complete recovery of RRP size. In between protocols, a constant flow of external solution was applied to the cells. For PDBu experiments, sucrose applications were performed as usual, after which neurons were incubated with 1 µM PDBu (Merck Millipore, Darmstadt, Germany), and sucrose applications were repeated. The order of sucrose solutions was alternated between neurons to avoid systematic errors due to possible rundown of RRP size after multiple applications. Other sources for systematic errors were investigated and, when experimentally assessable, found to be small for 0.5M and lower: sucrose responses were compared in the absence and presence of 0.2 mM kynurenic acid (Sigma, St. Louis MO), and no effect of receptor saturation on release kinetics was found for sucrose concentrations of 0.5M (*Figure 3—figure supplement 3*). Receptor desensitization did not affect RRP size measurements with 0.5M sucrose in a previous study

(*Pyott and Rosenmund, 2002*). However, we could not investigate its effect on release kinetics, since cyclothiazide (CTZ), next to blocking AMPA receptor desensitization, also stimulates the presynaptic release machinery (*Diamond and Jahr, 1995*; *Bellingham and Walmsley, 1999*; *Ishikawa and Takahashi, 2001*). We did not detect any contribution of HS-induced non-receptor currents, since subtracting the small current remaining after blocking NMDA and AMPA currents by 50 µM AP5 (Ascent) and 10 µM DNQX (Tocris, Bristol, UK) had a negligible effect on the fitted model rates (*Figure 3—figure supplement 4*). Offline analysis of electrophysiology was performed using Clampfit v9.0 (Molecular Devices), Mini Analysis Program v6.0 (Synaptosoft, Decatur GA), Axograph X (Axograph Scientific, Berkeley CA), and custom-written software routines (*Source code 1*) in Matlab 7.10.0 or R2010a (Mathworks, Natick MA).

## Vesicle state model

We used a minimal vesicle state model with a similar scheme as proposed by *Weis et al. (1999)* for $Ca^{2+}$-dependent vesicle pool dynamics in the Calyx of Held, consisting of a depot pool of non-primed vesicles $D$, RRP with primed vesicles $R$ and a fused pool $F$. Our model differs from the Weis-model on three aspects: (1) we model fusion as an continuous process during hypertonic stimulation, whereas in the Weis-model this is modelled as a discrete event during action potential stimulation, (2) in our model the rate constant for priming $k_1$ is constant, and not $Ca^{2+}$ dependent as in the Weis-model, since we use $Ca^{2+}$-independent stimuli to evoke release, and (3) opposed to Weis-model our model has a finite $D$ pool. This allowed us, in contrast to other pool models, to model synaptic responses to hypertonic sucrose, the relation between RRP depletion and release kinetics, and RRP replenishment during HS-stimulation.

Vesicle dynamics for the vesicles in the depot pool $D$ and the readily releasable pool $R$ are described by two-coupled differential equations

$$\frac{dD}{dt} = -k_1 D + k_{-1} R, \tag{7}$$

$$\frac{dR}{dt} = k_1 D - (k_{-1} + k_2) R, \tag{8}$$

with $k_{-1}$ and $k_2$ the rate constants for unpriming and fusion, respectively (*Figure 1B*). To compensate for leak of vesicles from the system due to spontaneous release, we would need an extra term in *Equation (7)* to refill $D$. However, since we assume the spontaneous release rate before sucrose stimulation to be negligibly small compared to the other rates, we can neglect the refill term in *Equation (7)*. *Equation (7)* was included to account for depletion of the depot pool during long or repetitive HS stimulation. However, for the durations of the HS stimulations used in this paper, depletion of $D$ was small and responses could be fitted with the priming rate $k_1 D$ being treated as a constant (see fitting procedures). For convenience, the pool sizes are expressed in nC instead of vesicles. In this version of the model, we did not include release sites since this would introduce an extra fit parameter, whereas such an extended model is mathematically equivalent (if immediate availability and recycling of release sites is assumed; see below). The RRP size at steady state is the result of a dynamic equilibrium between priming, unpriming, and fusion (*Weis et al., 1999*), and can be obtained from *Equation (8)* under the assumption of $dR/dt = 0$,

$$R_\infty = \frac{k_1 D}{k_{-1} + k_2}. \tag{9}$$

As mentioned above, for the purpose of determining the RRP size before stimulation, we assumed that $k_2$ was zero.

For simulation of synaptic responses to hypertonic stimulation, we assume that this form of stimulation selectively reduces the activation energy for fusion, and thus increases the release rate constant $k_2$ according to *Equation (4)*, without affecting upstream processes of fusion. Although solution exchange is very rapid (<0.5 s), the onset of a HS-evoked synaptic response starts with a delay with respect to the rise in hypertonicity, most likely due to compensatory mechanisms that initially successfully counteract this osmotic perturbation (see *Figure 2—figure supplement 2*). In addition, after the delay there is a smooth, rather than an abrupt transition to the evoked inward current. To capture these features, the time course of $k_2$ in response to sucrose is modelled as an expo-exponential

$$k_2(t) = k_{2,\text{max}} e^{-e^{-(t-t_0-t_{del})/\tau}} \quad (t \geq t_0),\tag{10}$$

with $t_0$ the time point of sucrose application, $t_{del}$ a constant which determines the delay of the onset of $k_2$ with respect to $t_0$, $\tau$ a time constant that sets the steepness of the rising phase, and $k_{2,\text{max}}$ the maximal value of $k_2(t)$ (*Figure 2B*). Each model parameter constrains the simulated HS-response in a specific way as shown in *Figure 2—figure supplement 3A* (absolute traces) and *Figure 2—figure supplement 3B* (traces scaled and aligned to peak). An increase in the priming rate constant $k_1$ or the depot pool $D$ both increases the total RRP and steady-state priming phase at the end of the response without affecting release kinetics. Decreasing the unpriming rate constant $k_{-1}$ increases the RRP, but without an effect on the steady-state priming phase. Increase of $t_{del}$ further delays the response but does not change its shape. Increase of the maximal fusion rate constant $k_{2,\text{max}}$ produces features that are typically observed experimentally when evoking post-synaptic responses with increasing levels of hypertonicity (*Figure 2A*), such as increase in peak amplitude, shorter the time to peak, and speed-up of the decay phase after the peak. Finally, decrease of $\tau$ speeds up the rise phase, increases the peak amplitude, but only mildly affects the decay phase after the peak. These characteristic effects allow the accurate estimation of the individual model parameters by fitting the vesicle state model to experimental HS-induced traces (see fitting procedures below).

## Analytical solution for hypertonic sucrose-induced release from a RRP without replenishment

By ignoring vesicle replenishment during HS-stimulation and the delayed onset of the HS-induced response, our vesicle state model can be simplified such that an analytical solution can be obtained that qualitatively captures the main features of HS-induced release. Release from a readily releasable pool $R$ without replenishment is given by

$$\frac{dR}{dt} = -k_2(t)R,\tag{11}$$

with $k_2(t)$ a release rate parameter that changes over time during the application of hypertonic sucrose with a time-course as described in *Equation (10)*. When neglecting the delayed onset of sucrose action, the time dependence of $k_2(t)$ can be approximated with a single exponential

$$k_2(t) = k_{2,\text{max}}\left(1 - e^{-\frac{t}{\tau}}\right)(t \geq 0),\tag{12}$$

with $k_{2,\text{max}}$ the maximal release rate, $\tau$ a time constant for the exponential time course of $k_2(t)$, and $t = 0$ the start of sucrose application. Solving *Equation (11)* analytically yields the following solution:

$$R(t) = R_0 e^{-k_{2,\text{max}}\left(\tau e^{-\frac{t}{\tau}}+t\right)+k_{2,\text{max}}\tau},\tag{13}$$

with $R_0 = R(0)$, the initial RRP size at the start of the stimulation. From this follows an exact expression for the fusion rate $k_2(t)R$:

$$\begin{aligned}\frac{dF}{dt} &= k_2(t)R\\ &= k_{2,\text{max}}\left(1 - e^{-\frac{t}{\tau}}\right)R_0 e^{-k_{2,\text{max}}\left(\tau e^{-\frac{t}{\tau}}+t\right)+k_{2,\text{max}}\tau}.\end{aligned}\tag{14}$$

After convolving fusion rates for different values of $k_{2,\text{max}}$ with an average mEPSC, postsynaptic current responses were obtained corresponding to different concentrations of hypertonic sucrose (*Figure 2—figure supplement 1*). These current responses display the typical characteristics as experimental responses, with increased peak release rates and shorter time-to-peak are observed for higher concentrations, but obviously do not reproduce the increased standing currents towards the end of depleting stimuli (0.5M or higher; *Figure 3A1*), because of the lack of replenishment in this model.

## Mathematical equivalent model with limited number of release sites

In our model described by *Equations (7)* and *(8)*, the number of release sites is not restricted. When we assume a fixed number of (instantaneously available) release sites $S$, *Equation (8)* transforms into

$$\frac{dR}{dt} = k_1 D(S - R) - (k_{-1} + k_2)R. \tag{15}$$

Here, the extra factor $(S - R)$ captures the idea that priming is hampered when fewer release sites are available for new vesicles to tether to. In this case, the steady-state RRP becomes

$$R_\infty = \frac{k_1 DS}{k_1 D + k_{-1} + k_2}. \tag{16}$$

If, as an approximation, we assume $k_1 D$ to be constant for the duration of the stimulation, *Equations (8)* and *(15)* and their respective steady-state RRP expressions *Equations (9)* and *(16)* are mathematically equivalent under the transformation $k_1 D \leftrightarrow (k_1 DS)_{sites}$ and $k_{-1} + k_2 \leftrightarrow (k_1 D + k_{-1} + k_2)_{sites}$. However, priming- and unpriming rate constants have different values in both systems and affect $R$ in a different manner.

## Vesicle replenishment

During hypertonic sucrose stimulation, vesicles are released from the RRP that consists of vesicles that were already primed at the onset of the stimulus $R_0$ and newly primed vesicles $R_{new}$. With $R = R_0 + R_{new}$, *Equation (8)* transforms into

$$\frac{d(R_0 + R_{new})}{dt} = k_1 D - (k_{-1} + k_2)(R_0 + R_{new}), \tag{17}$$

which can be separated in an expression for the depletion of $R_0$ and the replenishment of vesicles into $R_{new}$

$$\frac{dR_0}{dt} = -(k_{-1} + k_2)R_0, \tag{18}$$

$$\frac{dR_{new}}{dt} = k_1 D - (k_{-1} + k_2)R_{new}. \tag{19}$$

The postsynaptic current $I$ during the stimulus is given by the sum of the currents $I_{R0}$ and $I_{R_{new}}$, evoked by release from $R_0$ and $R_{new}$, respectively

$$\begin{aligned} I &= I_{R_0} + I_{R_{new}} \\ &= -k_2(t)(R_0 + R_{new}), \end{aligned} \tag{20}$$

with the minus sign correcting for the fact that we record inward currents but express $R$ in as positive charge (in nC).

Interestingly, in this reduced model it follows from *Equation (8)* that without a limited number of release sites and assuming $k_2 \approx 0$ in the absence of sucrose, recovery of the RRP after depletion is given by

$$R = (R_{end} - R_\infty)e^{-k_{-1}t} + R_\infty, \tag{21}$$

with $R_{end}$ the RRP size at the end of the depleting stimulus, $R_\infty$ the fully recovered RRP given by *Equation (9)*, and $1/k_{-1}$ the time constant for recovery.

## Analytical approximation for the relation between release kinetics and RRP depletion

The depleted RRP fraction is defined as the release during a hypertonic stimulus normalized to the steady state RRP size before the stimulation. If we assume that $R$ has an initial steady state value $R_i$ and is at a new steady state value $R_f$ at the end of the stimulus, the depleted RRP fraction can be expressed as

$$depleted\ RRP\ fraction = \frac{R_i - R_f}{R_i} = 1 - \frac{R_f}{R_i}. \tag{22}$$

Using *Equation (9)*, $R_i$ and $R_f$ are defined as

$$R_i = \frac{k_1 D}{k_{-1} + k_{2,0}}, \tag{23}$$

and

$$R_f = \frac{k_1 D_f}{k_{-1} + k_{2,max}}, \tag{24}$$

When we assume that $D$ is a large depot pool, with little effect on the size of $D$ from replenishment from $D$ to $R$ during a sucrose stimulus ($D_f \approx D_i$), and that the initial fusion ate before stimulation is negligibly small ($k_{2,0} \approx 0$), *Equation (22)* transforms into

$$\begin{aligned} Depleted \ RRP \ fraction &= 1 - \frac{(k_{-1} + k_{2,0})}{(k_{-1} + k_{2,max})} \frac{k_1 D_f}{k_1 D_i} \\ &\approx 1 - \frac{k_{-1}}{k_{-1} + k_{2,max}} \\ &= \frac{k_{2,max}}{k_{-1} + k_{2,max}}. \end{aligned} \tag{25}$$

This analytical approximation closely resembles the relation between $k_{2,max}$ and the depleted RRP fraction obtained with our model simulations using *Equations (7)*, *(8)*, and *(10)* (*Figure 4—figure supplement 1*).

## Fitting procedures and statistics

Fits were performed with an in-house developed analysis program in Matlab (*Source code 1*). The software reads Axon binary files (.abf), which can be loaded in batches.

When fitting the model to data, *Equations (8)* and *(10)* are numerically simulated using Matlab's *ode45* ordinary differential equation (ODE) solver. This one-step solver for non-stiff ODEs makes use of explicit Runge-Kutta methods of order 4 and 5 with a variable time step. Matlab's *odeset* structure to alter the ODE solver's properties, such as integration error and step size, is set to its default value. R is expressed in nC. The initial condition of the simulation is the steady-state solution of the model assuming $k_2 = 0$. During the initial fit of a trace, $k_1 D$ is taken constant and only *Equation (8)* is used. Subsequently, one can fit $D$ and $k_1$ separately to capture the decay in the refill phase, for instance during long HS-stimulations, by re-running the fitting procedure with all parameters (including RRP size and the product $k_1 D$) fixed, except for $D$ and $k_1$, using both *Equations (7)* and *(8)*. In this paper, $k_1 D$ is always obtained from the initial fit.

The data time span used for fitting is specified by the user, and is generally taken equal to the duration of the sucrose application, up to the time when the sucrose concentration starts to decay back to baseline. The solution for the $R$ state in this time window resulting from the ODE solver is subsequently interpolated at each measured time point within the fitting time window (typical sampling frequency 10 kHz) and the outcome is fed into a cost function, which calculates the sum of squared errors between model prediction and data for each iteration. When fitting multiple sucrose responses of a single cell simultaneously (e.g., 0.5M and 0.25M), the sum of squared errors is calculated separately for each concentration and subsequently added up. This cost function is used as input for the optimisation algorithms, all of which are contained in Matlab's Optimization Toolbox. The user has the option to choose between global (genetic algorithm or simulated annealing) and local (Nelder-Mead downhill simplex) methods. All methods are executed using default options, except for the lower and upper bounds of all parameters as used by the global search methods, which are set to $10^{-5}$ and $10^6$, respectively. The user can control the maximum number of iterations and function evaluations, both of which are by default set to 400 per fitted parameter. Once the global method has reached its stopping criterion at a certain point in parameter space, the local method takes over to search for the optimal set of parameters in the neighbourhood of this point. Quality of the fits was checked by visual comparison of the following features between the fitted curve and the experimental trace: (1) onset of fit, (2) peak amplitude and/or time-to-peak, (3) decay towards steady state phase, and (4) steady-state phase (refill) (*Figure 3—figure supplement 5B*). When the deviation was too large, traces were fitted again with new initial conditions until no further improvement of the

fit was observed. Although the model consists of multiple free parameters, different features of the HS-induced traces are constrained by different parameters in the model (*Figure 2—figure supplement 3*) and vice versa. The RRP size, and thus the ratio of $k_1 D$ and $k_{-1}$, is constrained by the charge transfer during the peak. In addition, $k_1 D$ is constrained by the steady state current after the peak, which then also constrains $k_{-1}$ via the RRP size and *Equation (9)*. Note that the RRP itself is not a fit parameter, and that the fit procedure optimizes $k_1 D$ and $k_{-1}$ to get the best fit of the experimental trace. *Equation (9)* is then used to calculate the RRP post-hoc. $t_{del}$ is constrained by the delay of the onset of the response. Peak amplitude in combination with steepness of the rise phase constrains $\tau$, and peak amplitude in combination with the decay phase after the peak constrains $k_{2,max}$. Simulations show that the fit method can indeed robustly discriminate between the effects of different model parameters on the shape of the sucrose response, that is, changes in one model parameter are reliably detected with the other model parameters being invariant (*Figure 3—figure supplement 5A,C*; *Figure 3—source data 2*). In addition, random examples of experimentally obtained responses to 0.3M and 0.5M sucrose in the absence and presence of the phorbol ester (PDBu) show that this method provides a close fit for almost all traces (*Figure 6—figure supplements 1, 2*).

The activation energy as a function of sucrose concentration as shown in *Figure 3D* was fitted with a mono-exponential function of the form $\Delta E(M) = ae^{-b \cdot M} + c$, with $M$ the sucrose concentration in molar, using Matlab's built-in Curve Fitting Tool. Fits of $k_{2,max}$ as a function of sucrose concentration in *Figure 3C* were obtained by transformation of the fitted function in *Figure 3D*, using *Equation (5)*. As log-transforming symmetrical error bars in the release rate domain results in asymmetric error bars in the energy domain, we used the largest error of the two for plotting the SEM of fitted activation energy. Data shown in Figure are mean ± SEM. In addition, bootstrap analysis was performed to estimate statistical errors and confidence intervals for the distributions of the mean values of all fitted parameters. We applied the nonparametric bootstrap method (i.e., resampling the original data) using the 'bootstrap' function from MATLAB's statistics toolbox with default options. The size of the original data sets used to constitute the bootstrap sample is equal to the number of observations per parameter (n), as given in the figure tables. For each parameter, we bootstrapped 10,000 sample means, and subsequently calculated the mean value, the standard deviation (std) and the 95% confidence interval (95% CI) of the distributions of these sample means. For the combined effect of PDBu and sucrose on $k_{2,max}$ we also calculated 95% CI for the absolute change in $k_{2,max}$ (*Figure 6D*). Values used for model parameters and fit parameters in the figures and results from bootstrap analysis are given in the supplemental tables provided for each figure.

## Acknowledgements

We would like to thank E Neher for discussions and valuable comments on the manuscript, J Broeke for contributing to software development, M Xue for providing data for reanalysis, D Schut for technical assistance and C van der Meer and B Tersteeg for animal breeding. This work was supported by Netherlands Organisation for Scientific Research (NWO), (CLS2007 project 635.100.020, Complexity project 645.000.003 and TOP project 91207032 to LNC, and MV) and the EU (SynSys Health-F4-2010-242167, and Eurospin Health-F2-2009-241498, to MV and JBS).

## Additional information

### Competing interests

CR: Reviewing editor, *eLife*. The other authors declare that no competing interests exist.

### Funding

| Funder | Grant reference | Author |
| --- | --- | --- |
| Nederlandse Organisatie voor Wetenschappelijk Onderzoek | CLS2007 project 635100020, Complexity project 645.000.003, TOP project 91207032 | Sebastiaan Schotten, Marieke Meijer, Matthijs Verhage, Lennart Niels Cornelisse |
| Fourth Framework Programme (European Union Fourth Framework Programme) | Synsys Health-F4-2010-242167, Eurospin Health-F2-2009-241498 | Nils Brose, Jakob B Sørensen, Matthijs Verhage |

| Funder | Grant reference | Author |
| --- | --- | --- |
| Nederlandse Organisatie voor Wetenschappelijk Onderzoek | Complexity project 645000003 | Lennart Niels Cornelisse |
| European Commission | Second Framework Programme: Eurospin Health-F2-2009-241498 | Nils Brose, Jakob B Sørensen, Matthijs Verhage |
| Nederlandse Organisatie voor Wetenschappelijk Onderzoek | TOP project 91207032 | Matthijs Verhage, Lennart Niels Cornelisse |

The funders had no role in study design, data collection and interpretation, or the decision to submit the work for publication.

### Author contributions

SS, Modelling, Analysis and interpretation of data, Drafting or revising the article; MM, VH, LM, LK, MV, MR, Acquisition of data, Analysis and interpretation of data; AMW, JBS, MV, Conception and design, Analysis and interpretation of data, Drafting or revising the article; NB, CR, Conception and design, Drafting or revising the article; LNC, Modelling, Conception and design, Acquisition of data, Analysis and interpretation of data, Drafting or revising the article

## Additional files

### Supplementary file

• Source code 1. Custom software to analyze HS-induced postsynaptic currents written in MATLAB (only compatible with MATLAB R2013 or older). Instructions for how to use the program are in the readme file. Use on a Mac or Linux system requires specification of the location of the poi_library when asked for by the program.

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
