## [Decision Letter]

Thank you for sending your work entitled “Additive effects on the energy barrier
for synaptic vesicle fusion cause supralinear effects on the vesicle fusion rate”
for consideration at *eLife*. Your article has been favorably evaluated
by Randy Schekman (Senior editor), a Reviewing Editor, and two reviewers, one of whom,
Frederic Pincet, has agreed to share his identity.

The Reviewing editor and the reviewers discussed their comments before we reached this
decision, and the Reviewing editor has assembled the following comments to help you
prepare a revised submission.

In this combined experimental and modelling study, the authors develop a kinetic model
in which they describe the RRP release kinetics, and then use this model to show
experimentally (using sucrose-evoked release at hippocampal autapses in culture) that
the fusion rate is supralinearly dependent on the energy barrier to fusion. The
consensus of the reviewers is that the idea that there is an additive/multiplicative
relationship between energy barrier and the fusion rate is an attractive one, and the
agreement between experiment and model is intriguing. However, the reviewers expressed
some serious concerns about the manuscript, which are described in detail below. To
address these, the authors need to perform additional control experiments, provide more
explanation about the details of the model and how the model was derived from the
experimental data, and place their results better in the context of the literature.

1) The model is briefly described in the second section of the Results (Minimal vesicle
state model). It is basically a chemical reaction between a depot vesicle pool, a
readily releasable pool (RRP) and fused vesicles. However, more clarification is
required regarding the following points:

A) The authors should explain better which features of the data/traces constrain which
parameters in the model, alone or in combination, during the fitting procedure. This is
required to better understand the reliability of the fitting procedure, how the
parameter values are identified by it, and therefore how the conclusions of the
manuscript are actually reached. Additional supplemental figures might be helpful in
this regard.

B) How different is this model from several published pool models, including an example
cited in the paper (Weis, 1999)? The authors should clarify which parts of the model the
authors specifically derive for the case of the RRP are actually novel.

C) How are the various parameters derived from the experimental data? There is a brief
description in the supplementary materials, but, for instance, it is not clear whether
(and how) k_1_, k_-1_, D and R were separately obtained from the
steady state before fusion and the response during fusion (e.g. provide an example using
supplementary information if necessary).

D) It seems that the third, fourth and fifth sections of the Results are meant to
validate the model by showing that R (amount of vesicles in the RRP) predicted by the
model is consistent with what is obtained experimentally. However, quantitative
comparisons with experiments are lacking. The authors have to use experimental values
(their own or published ones) to prove their point. This is true for R, but also for the
various rates in [Disp-formula equ6] and [Disp-formula equ7]. Also, why do the parameters change
from one table to the next?

E) In the fifth section of the Results (“Modulation energy barrier by genetic and
biochemical perturbations”) and corresponding Figure 5, the authors should provide more explanation for how they obtained
these curves. How do they define the peak release rate?

F) The authors should provide more discussion about whether the rates and sizes of the
various pools derived by their model are reasonable in light of previously estimated
values for these parameters.

2) The authors aim to show supralinear dependence of the fusion rate with the activation
energy for fusion. The idea that additive effects on energy barrier causes supralinear
effects is not new, and is expected for any thermally activated system. This has been
known for many decades (see H. A. Kramers, Physica (Amsterdam) 7, 284, 1940 and P.
Hänggi, P. Talkner, and M. Borkovec, Rev. Mod. Phys. 62, 251 1990), and recently
revisited by Evans and others (e.g. E. Evans and P. M. Williams, in Physics of
Bio-Molecules and Cells, edited by F. Julicher, P. Ormos, F. David, and H. Flyvbjerg
Springer-Verlag, Berlin, Germany, 2002, p. 145). So, the primary goal here is to show
that, indeed, activation energy is a key element that controls the kinetics of fusion.
This raises the following concerns:

A) It appears that [Disp-formula equ1] is valid
only in vacuum where quantum vibrations can be connected with thermal agitation. In the
present case, viscosity makes the system overdamped and the prefactor is more of the
order of 10^7-10^10 s^-1 (see references above). Currently it is
∼10^13 s^-1. Should not all activation energy values be shifted by
∼6kBT?

B) There is a sort of loophole in the current reasoning presented in the manuscript. The
authors start from a non-linear equation ([Disp-formula equ1]), use this equation to deduce activation energy from the fusion
rate and claim that the fusion rate varies non-linearly with the activation energy. The
only result that suggests that the supra linearity is real is that presented in Figure 6 where the presence of PDBu increases the
fusion rate and this increase can be explained by a constant shift of the activation
energy barrier for all hypertonic stimulations. This strongly suggests that the
supra-linear model is correct in this case. However, this is not true anymore with
Complexin for which the shift of the activation energy barrier is dependent on the level
of hypertonic stimulation. This difference in behavior should be specifically
discussed.

3) In the recordings of sucrose-evoked EPSCs, it is unclear how the authors dealt with
nonlinearities of the postsynaptic response, such as saturation or desensitization of
postsynaptic receptors. Control experiments with kynurenic acid + cyclothiazide
would help address this issue.

4) Related to the above point, why do the traces of HS responses become less noisy in
the high sucrose concentration limit? This could indeed be due to desensitization or
saturation. Or is this simply due to a different number of traces used for averaging?
Clearly, this point must be rigorously addressed.

5) Systematic and statistical errors for the estimated rates should be determined.
Statistical errors, for example, could be easily obtained by bootstrap analysis.

6) The authors claim several times that the sucrose evoked release is
Ca^2+-^independent, but do we really know this? The best solution
would be to perform control experiments in the absence of extracellular
Ca^2+^. At the very least, this point should be phrased and discussed
more carefully, referring to the relevant literature.

7) The exchange time for the sucrose application should be quantified a bit better. It
is unclear how non-instantaneous and non-uniform exchange will affect the
conclusions.

8) An interesting implication of the present work is that the nonlinear relation between
activation energy and rate may underlie the cooperativity in the Ca^2+^
dependence of release. This is only briefly discussed in the paper. The manuscript would
benefit from an expansion of this interesting aspect.

---

## [Author Response]

*1) The model is briefly described in the second section of the Results (Minimal
vesicle state model). It is basically a chemical reaction between a depot vesicle
pool, a readily releasable pool (RRP) and fused vesicles. However, more clarification
is required regarding the following points*:

*A) The authors should explain better which features of the data/traces constrain
which parameters in the model, alone or in combination, during the fitting procedure.
This is required to better understand the reliability of the fitting procedure, how
the parameter values are identified by it, and therefore how the conclusions of the
manuscript are actually reached. Additional supplemental figures might be helpful in
this regard*.

We agree that a better explanation is in fact required. We created four supplemental
figures and added detailed descriptions in the Materials and methods. In Figure 2—figure supplement 3 we added how
each model parameter affects the shape of the hypertonic sucrose (HS)-induced response
in a specific manner. This is further explained in the subsection headed “Vesicle
state model”. In Figure 3—figure supplement 5 and in the subsection “Fitting procedures and
statistics”, we explain how our fit method reliably discriminates between the
contributions of different model parameters to the HS-induced response. Finally, in
Figure 6—figure supplement 1 and Figure 6—figure supplement 2, we display random examples (10 per condition) of synaptic responses induced
by 0.3 or 0.5M sucrose in the presence or absence of PDBu to show how well these traces
are fitted by our model, which is also mentioned in “Fitting procedures and
statistics”.

B) How different is this model from several published pool models, including an
example cited in the paper (Weis, 1999)? The authors should clarify which parts of
the model the authors specifically derive for the case of the RRP are actually
novel.

We added a more elaborate description in the Materials and methods of how our model
differentiates from the Weis et al. model (please see the subsection “Vesicle
state model”). The unique/novel features of our model are that for the first time
synaptic responses to hypertonic sucrose are modelled, and the relation between RRP
depletion and release kinetics, and RRP replenishment during HS-stimulation. This is now
added to the aforementioned subsection. Our model also differs from other pool models in
the number of primed pools that is assumed, 1 pool vs 2 (Walter, 2013, Wolfel, 2007) or
3 pools (Voets, 1999), and the trigger for release that is simulated, hypertonic sucrose
vs flash-Ca^2+^ (Walter, 2013, Wolfel, 2007, Voets, 1999).

*C) How are the various parameters derived from the experimental data? There is a
brief description in the supplementary materials, but, for instance, it is not clear
whether (and how) k*_*1*_*,
k*_*-1*_
*, D and R were separately obtained from the steady state before fusion and the
response during fusion (e.g. provide an example using supplementary information if
necessary).*

We agree that the description of how the parameters are derived from experimental data
has not been optimal. We have added an explanation in the Materials and methods. As
described above (point 1A) and shown in Figure 2—figure supplement 3, different parameters constrain unique features
of the HS-induced synaptic responses. We use this the other way around to constrain the
model parameters when fitting the experimental traces as explained in the
“Fitting procedures and statistics” subsection and shown in Figure 3—figure supplement 5. All
parameters are obtained fitting the response during fusion. R at steady state is
obtained from the ratio of the fitted parameters k_1_D/k_-1_. Although
there is an option in our software to fit k_1_ and D separately to account for
decay of refill phase, for instance during long HS-stimulations, we used here the option
to fit product k_1_D as a constant (priming rate). This is more clearly
explained in the subsections “Vesicle state model” and “Fitting
procedures and statistics”.

*D) It seems that the third, fourth and fifth sections of the Results are meant
to validate the model by showing that R (amount of vesicles in the RRP) predicted by
the model is consistent with what is obtained experimentally. However, quantitative
comparisons with experiments are lacking. The authors have to use experimental values
(their own or published ones) to prove their point. This is true for R, but also for
the various rates in*
[Disp-formula equ6] and [Disp-formula equ7]*. Also, why do the parameters
change from one table to the next?*

The reviewers are right. Quantitative comparisons with experiments are lacking and
should have been included to validate the model. We have now included comparisons of our
fitted parameters with estimated parameter values in previous studies, except for
k_2.max_ and the activation energy associated with HS-induced release since
we are the first to quantify these with our method. k_1_D is compared with
published priming rates (subsection headed “Assessing RRP size and release rate
constants”). RRP is compared with pool sizes from 7 different studies. We have
added a derivation for RRP recovery after depletion in the Materials and methods section
([Disp-formula equ21]) to compare the inverse
of k_-1_ with previously published RRP recovery time constants in autapses. As
discussed in the text, all our parameter estimates are in the range of published values.
Variation in the parameters between different tables arises from the fact that they are
obtained from different experiments, performed at different points in time and in
different labs. Hence, this is general variation between experiments.

*E) In the fifth section of the Results (“Modulation energy barrier by
genetic and biochemical perturbations”) and corresponding*
Figure 5*, the authors
should provide more explanation for how they obtained these curves. How do they
define the peak release rate?*

We agree and have now provided more explanation, both in the main text (in the
subsection entitled “Modulation of the activation energy for fusion by genetic
and biochemical perturbations”) and in the figure legend of Figure 5. Peak release rate is defined as the release rate at the
peak of a HS-induced response. We plotted the reported peak release rates and
corresponding depleted RRP fractions for different perturbations in one graph in Figure 5. The predicted curve in Figure 5 is obtained by plotting the peak release
rates and depleted RRP fractions obtained from different simulations of HS-induced
responses with only the model parameter k_2max_ varied.

*F) The authors should provide more discussion about whether the rates and sizes
of the various pools derived by their model are reasonable in light of previously
estimated values for these parameters*.

We agree, see point 1D.

*2) The authors aim to show supralinear dependence of the fusion rate with the
activation energy for fusion. The idea that additive effects on energy barrier causes
supralinear effects is not new, and is expected for any thermally activated system.
This has been known for many decades (see H. A. Kramers, Physica (Amsterdam) 7, 284,
1940 and P. Hänggi, P. Talkner, and M. Borkovec, Rev. Mod. Phys. 62, 251
1990), and recently revisited by Evans and others (e.g. E. Evans and P. M. Williams,
in Physics of Bio-Molecules and Cells, edited by F. Julicher, P. Ormos, F. David, and
H. Flyvbjerg Springer-Verlag, Berlin, Germany, 2002, p. 145). So, the primary goal
here is to show that, indeed, activation energy is a key element that controls the
kinetics of fusion. This raises the following concerns*:

*A) It appears that*
[Disp-formula equ1]
*is valid only in vacuum where quantum vibrations can be connected with thermal
agitation. In the present case, viscosity makes the system overdamped and the
prefactor is more of the order of 10^7-10^10 s^-1 (see
references above). Currently it is ∼10^13 s^-1. Should not all
activation energy values be shifted by ∼6kBT?*

We completely agree with the reviewers. [Disp-formula equ1] is in fact not the right equation to apply. The assumption of
thermally driven dissociation of the activated complex in a quantum oscillator model is
not valid for our reaction (which occurs in solution and involves complex molecular
rearrangements of proteins and lipids). The consequence is that the previous values of
the exponential prefactor were an overestimation, as the reviewers correctly pointed
out. To circumvent the necessity of investigating assumptions underlying the derivation
of the Eyring equation (which may not be valid in our system), in the revised version,
we decided to describe the rate constants based on the empirical Arrhenius equation,
which differs from the Eyring equation by containing an empirical prefactor (A). A
consequence of this new notation is that we now consider relative changes of the
activation energy (E_a_) for fusion, not absolute values, at least until the
Discussion. We have now plotted energy differences with respect to the activation energy
for fusion at rest, which also allows us to directly show the estimated consequences of
hypertonic solutions. In this way, we can still substantiate all our main claims, but
the switch to an empirical equation leads to altered labels on most graphs and [Disp-formula equ1]-[Disp-formula equ5]. We thank the reviewers for pointing out this crucial flaw in our
reasoning.

In addition, using the Arrhenius equation makes it clearer that an increase in the
reaction rate constant can result either from an increase in the pre-exponential factor
(A), or from a decrease of the energy barrier (E_a_), or both. In our
manuscript we now discuss these possibilities explicitly and provide arguments that the
effect of hypertonicity is most likely due to a reduction in the energy barrier (please
see the subsections “Sucrose stimulation reflects a decrease in the activation
energy for fusion” and “The Arrhenius equation infers the activation
energy for synaptic vesicle fusion”). We have also added a sentence emphasizing
that changes in the pre-exponential factor will also contribute to changes in the fusion
rate.

*B) There is a sort of loophole in the current reasoning presented in the
manuscript. The authors start from a non-linear equation (*[Disp-formula equ1]*), use
this equation to deduce activation energy from the fusion rate and claim that the
fusion rate varies non-linearly with the activation energy. The only result that
suggests that the supra linearity is real is that presented in*
Figure 6
*where the presence of PDBu increases the fusion rate and this increase can be
explained by a constant shift of the activation energy barrier for all hypertonic
stimulations. This strongly suggests that the supra-linear model is correct in this
case. However, this is not true anymore with Complexin for which the shift of the
activation energy barrier is dependent on the level of hypertonic stimulation. This
difference in behavior should be specifically discussed*.

We acknowledge this flaw in our previous reasoning. We have removed this and now start
with stating explicitly that: “direct measurements of the exact contributions of
different molecular events inside living nerve terminals to the activation energy for SV
fusion are not possible” (please see the Introduction and the subsection entitled
“The Arrhenius equation infers the activation energy for synaptic vesicle
fusion”). Subsequently, we argue that according to the Arrhenius equation a class
of modulations of synaptic release might exist, which will result in supralinear effects
on synaptic release rate through an additive effect on the activation energy. The effect
of PDBu is indeed in line with this (as noted by the reviewers), whereas the effect of
complexin appears to be more complex. As the reviewers suggest, we have discussed this
in the revised manuscript (“Supralinear modulation of release kinetics by Phorbol
esters and Complexins through additive effects on the activation energy”).
Several previous observations in literature provide leads (clamping a secondary
Ca^2+-^sensor for spontaneous and asynchronous release, rendering the
synapse more sensitive to spontaneous Ca^2+^ fluctuations, changing the
cooperativity of exocytosis).

*3) In the recordings of sucrose-evoked EPSCs, it is unclear how the authors
dealt with nonlinearities of the postsynaptic response, such as saturation or
desensitization of postsynaptic receptors. Control experiments with kynurenic acid
+ cyclothiazide would help address this issue*.

We agree with the reviewers that the effect of receptor desensitization or saturation on
sucrose-evoked EPSCs have been underexposed in our manuscript. These issues were
extensively studied by Pyott et. al., 2002 using kynurenic acid + cyclothiazide.
No effect was found on RRP measurements with 0.5M. We refer now to this paper. In
addition, we performed new experiments and tested whether AMPA receptor saturation
affected the kinetics of synaptic responses to 0.5 and 0.75M. We confirmed that
saturation does not affect measurements with 0.5M. However, for 0.75M the release rate
constant was about 30% faster in the presence of KYN (see new Figure 3—figure supplement 3). We conclude that
quantifications of model parameters obtained from responses to 0.75M and higher should
be interpreted with caution (see the subsection entitled “Assessing RRP size and
release rate constants”). However, this does not change any of our main
conclusions.

Several studies report that, in addition to blocking AMPA receptor desensitization, CTZ
stimulates the presynaptic release machinery (Bellingham, 1999, Diamond, 1995, Ishikawa,
2001). Therefore, we could not use this drug to test a potential effect of receptor
desensitization, as suggested by the reviewers. However, when examining noise levels
(see point 4 below), we concluded that desensitization was negligible for responses to
concentrations up to 0.5M. Since the experiments with PDBu or Complexin were performed
with 0.5M and lower, we conclude that these measurements were not affected by receptor
saturation or desensitization. This is added to the revised manuscript.

*4) Related to the above point, why do the traces of HS responses become less
noisy in the high sucrose concentration limit? This could indeed be due to
desensitization or saturation. Or is this simply due to a different number of traces
used for averaging? Clearly, this point must be rigorously addressed*.

All traces are single responses (not averaged), recorded with the same filter settings.
We analyzed the noise levels on all traces and found indeed that responses to sucrose
concentrations beyond 0.5M tend to have lower noise levels. In view of point 3 above, we
interpreted this as an effect of saturation and/or desensitization. This point is now
addressed in the text where we state that quantifications of model parameters obtained
from responses to 0.75M and higher should be interpreted with caution. We thank the
reviewers for pointing this out.

5) Systematic and statistical errors for the estimated rates should be
determined. Statistical errors, for example, could be easily obtained by bootstrap
analysis.

We have performed experiments with KYN to determine systematic errors due to receptor
saturation (see point 3, Figure 3—figure supplement 3). In addition we have performed experiments with glutamate
receptor blockers DNQX and AP5 to determine the systematic error due the contribution of
non-receptor currents to HS-induced responses, and found that this effect was negligible
(Figure 3—figure supplement 4).
Bootstrap analysis was performed for all experiments to determine 95% confidence
intervals. Results are now given in the supplementary tables. For the effect of PDBu and
complexin on the fusion rate constant, we calculated the 95% confidence intervals for
the mean difference in k_2,max_ between the experimental (PDBu or CpxKO) and
the control (no PDBu or Cpx WT) group and plotted these as error bars in Figures 6 and 7. All mean differences were
within the calculated 95% confidence intervals.

*6) The authors claim several times that the sucrose evoked release is
Ca*^*2+-*^*independent, but do we
really know this? The best solution would be to perform control experiments in the
absence of extracellular Ca*^*2+*^*. At
the very least, this point should be phrased and discussed more carefully, referring
to the relevant literature*.

The Ca^2+^-dependency of HS-induced responses has been extensively
studied, for instance in Rosenmund and Stevens in Neuron (1996), showing that neither
buffering intracellular Ca^2+^ by BAPTA nor blocking
Ca^2+^ influx using CdCl_2_ had an effect (Rosenmund, 1996).
Several other extensive studies are already available. We discuss this now more
carefully in the text with a reference to this paper (in the subsection headed
“Minimal vesicle state model for synaptic vesicle release”).

*7) The exchange time for the sucrose application should be quantified a bit
better. It is unclear how non-instantaneous and non-uniform exchange will affect the
conclusions*.

To address this point we have performed open-tip experiments with 0.5M and 1M sucrose
(Figure 2—figure supplement 2). We
show that the solution exchange is instantaneous (within 0.4 seconds after switching
barrels), compared to the induced postsynaptic currents, which respond with a delay of
1.1 (1M)-1.6s (0.25M), and therefore will not affect the conclusions in this paper. This
is discussed in the first paragraph of the Materials and methods section.

*8) An interesting implication of the present work is that the nonlinear relation
between activation energy and rate may underlie the cooperativity in the
Ca*^*2+*^
*dependence of release. This is only briefly discussed in the paper. The
manuscript would benefit from an expansion of this interesting aspect*.

We agree with the reviewers that this is an interesting implication of our work.
Therefore we expanded the discussion with a derivation of the allosteric model for
Ca^2+^ dependence of vesicle release within the framework of the
paper. We show that the supralinear relation between the intracellular
Ca^2+^ concentration and the fusion rate follows directly from eq.
(Bellingham, 1999), when assuming that the Ca^2+^ sensor reduces the
activation energy for fusion with a fixed amount
*ΔE*_*Ca*_ for each
Ca^2+^ ion binding. This is illustrated in a new figure (Figure 8).